# FEDERATED GENERALIZATION VIA INFORMATION-THEORETIC DISTRIBUTION DIVERSIFICATION

## ABSTRACT

Federated Learning (FL) has surged in prominence due to its capability of collaborative model training without direct data sharing. However, the vast disparity in local data distributions among clients, often termed the non-Independent Identically Distributed (non-IID) challenge, poses a significant hurdle to FL's generalization efficacy. The scenario becomes even more complex when not all clients participate in the training process, a common occurrence due to unstable network connections or limited computational capacities. This can greatly complicate the assessment of the trained models' generalization abilities. While a plethora of recent studies has centered on the generalization gap pertaining to unseen data from participating clients with diverse distributions, the divergence between the training distributions of participating clients and the testing distributions of non-participating ones has been largely overlooked. In response, our paper unveils an information-theoretic generalization framework for FL. Specifically, it quantifies generalization errors by evaluating the information entropy of local distributions and discerning discrepancies across these distributions. Inspired by our deduced generalization bounds, we introduce a weighted aggregation approach and a duo of client selection strategies. These innovations aim to bolster FL's generalization prowess by encompassing a more varied set of client data distributions. Our extensive empirical evaluations reaffirm the potency of our proposed methods, aligning seamlessly with our theoretical construct.

## 1 INTRODUCTION

Federated Learning (FL) offers a collaborative paradigm to train a shared global model across distributed clients, ensuring data privacy by eliminating the need for direct data transfers (Zhu et al., 2021; McMahan et al., 2017). However, the inherent heterogeneity among clients—often due to their distinct operational environments—generates non-Independent and Identically Distributed (non-IID) scenario (Hu et al., 2023; Zhu et al., 2021; Zhao et al., 2018). This distinctness complicates the assessment of FL's generalization capabilities, setting it apart from traditional centralized learning (Yuan et al., 2021; Mohri et al., 2019).

While prevailing research on FL generalization predominantly concentrates on actively participating clients (Mohri et al., 2019; Qu et al., 2022). it offers a limited view: it addresses the model's adaptability to observed local distributions without reconciling the divergence between observed data distributions in actively participating clients and unobserved data distributions from passive unparticipating clients. In real-world settings, numerous clients might remain detached from the training process, due to unstable network connectivity or other constraints (Hu et al., 2023; Lim et al., 2020). For example, in cross-device federated edge learning, IoT devices, although reliant on robust deep learning models for edge inferences (Wang et al., 2023; Pan et al., 2022), often abstain from FL owing to computational and communicative constraints (Tak and Cherkaoui, 2020). This landscape raises a pivotal question: Can models, honed by active participants, cater effectively to passive clients that do not participate in the training process?

Some recent endeavours strive to demystify this by quantifying the performance disparity between models tested on active versus passive clients, proposing a dual-level framework to evaluate both out-of-sample and out-of-distribution generalization gaps, grounded in the premise that client data distributions stem from a meta-distribution (Hu et al., 2023; Yuan et al., 2021). Yet, the approach

by Yuan et al. (2021) primarily sketches the contours of the FL generalization gap, substantiating it with empirical evidence sans a comprehensive theoretical underpinning. Conversely, the approach byy Hu et al. (2023) accentuates theoretical insights on meta-distribution-based errors but leave practical algorithmic solutions largely unexplored, particularly those that might uplift the global model's adaptability for passive clients.

**Motivation.** Traditional Federated Learning (FL) typically deploys a global model for actively participating clients, assuming an alignment between training and testing data distributions. Our investigation diverges from this standard, spotlighting the global model's out-of-distribution generalization capabilities when trained by participating clients. In essence, we seek to cultivate a model via FL across active clients that reliably serves even the passive ones. Specifically, a novel contribution of our paper is the introduction of the self-information weighted expected risk – a metric to gauge model generalization. Our hypothesis is grounded in the notion that a model exhibiting proficiency with low-probability examples from training distributions might demonstrate adaptability to unfamiliar testing distributions. Such examples are anticipated to hold a greater significance in these unseen datasets. Operationalizing this concept within FL, our framework leverages the self-information of examples to craft a generalization boundary. Delving into the resulting entropy-aware and distribution discrepancy-aware generalization disparities, a revelation emerges: data sources tend to manifest informational redundancy. This implies that certain data sources, characterized by diminished informational weight, can be seamlessly supplanted by others. Essentially, only a fraction of the client base holds substantial sway over FL's generalization. This redundancy is especially pronounced in AIoT settings, where a myriad of edge devices—often operating in overlapping zones—partake in FL. Consider drones, serving as FL clients, amassing spatial data for model training. Drones constrained by limited flight spans might be rendered redundant due to their shared operational territories with other drones (Wang et al., 2021). Informed by these insights, our paper further proposes strategies to elevate the generalization prowess of FL.

**Contributions.** Our paper presents several key contributions.

- We introduce a novel theoretical framework to scrutinize the generalization error in FL. Distinctly, our approach shines a light on the "participation gap", an aspect largely glossed over in preceding research. This framework adeptly harnesses the information entropy of data sources, coupled with the distribution variances among them, to provide a more refined insight into the generalization potential of models.

- Drawing from our theoretical results, we champion a weighted aggregation approach alongside a duo of client selection methodologies. These are designed to amplify the generalization prowess of FL.

- The empirical evaluations using three widely-referenced datasets underscore the efficacy of our methods, consistently eclipsing benchmarks.

## 2 RELATED WORK

Data heterogeneity is a major challenge in federated learning (Zhu et al., 2021; Reisizadeh et al., 2020; Ma et al., 2022). Despite numerous studies investigating the generalization error in the presence of data heterogeneity (Mohri et al., 2019; Qu et al., 2022; Caldarola et al., 2022), most of these have focused only on scenarios where a global model is trained on distributed data sources and tested on unseen data from these sources. While Yuan et al. (2021) and Hu et al. (2023) also consider this generalization problem in FL, they do not account for local distribution characteristics nor propose methods to enhance generalization performance. In contrast, our work is motivated by the need to design algorithms providing good generalization performance on passive clients (that unparticipate in the training process) with unknown data distributions. Our framework takes into account the local distribution property and provides methods for improving generalization performance in this setting.

There have been several studies that propose information-theoretic generalization analyses for FL (Yagli et al., 2020; Barnes et al., 2022; Sefidgaran et al., 2022). For instance, Yagli et al. (2020) developed a framework for generalization bounds that also accounts for privacy leakage in FL. Barnes et al. (2022) presented generalization bounds for FL problems with Bregman divergence or Lipschitz continuous losses. However, these works focus on bounding the generalization error via the mutual information between models and samples, ignoring the information stored in data sources. And

they only consider the IID generalization error, with the assumption that the source distribution is identical to the target distribution. Compared with Federated Domain Generalization (FedDG) (Zhang et al., 2023; Nguyen et al., 2022b) dealing with the challenge of domain distribution shift under FL settings, our focus is solely on a specific collection of all the data sources in FL. Our objective is to train a global model from seen distributions of participating clients and generalize it to other unseen distributions of unparticipating clients. Additionally, it is commonly assumed that each domain contains equal information and cannot represent each other in FedDG. However, we consider the presence of information redundancy among data sources and only a subset of clients contributes to the generalization of FL. Besides, most DG studies concentrate on learning an invariant representation across different domains (Nguyen et al., 2022a; Li et al., 2022; Lu et al., 2022; Muandet et al., 2013). Conversely, our focus is to design proper weighting aggregation and client selection methods to mitigate the generalization error in FL.

## 3 THEORETICAL FRAMEWORK

Similar to previous studies (Yuan et al., 2021; Hu et al., 2023), we model each data source as a random variable with its corresponding distribution. In order to better evaluate the generalization performance of FL, we focus on the gap between the information-theoretic global objective in FL and the self-information weighted expected risk based on the joint distribution of all the data sources. To conserve space, we have provided proofs for all our theorems in the appendix.

### 3.1 PRELIMINARIES

Let the sample space $\mathcal{Z} = \mathcal{X} \times \mathcal{Y}$ be the set of all the possible outcomes $z = (x, y)$ (e.g. image-label pairs) focused on in this paper, where $\mathcal{X}$ is the feature space and $\mathcal{Y}$ is the label space. Let $\mathcal{I}$ be the index set of all the possible clients. The total number of all possible clients in $\mathcal{I}$ is $|\mathcal{I}| = N$ and it is possibly infinite. We assume that only clients in the finite subset $\mathcal{I}_p$ of $\mathcal{I}$ participating in FL practically and the number of these clients is $|\mathcal{I}_p| = M$. Following Hu et al. (2023), $N$ is commonly larger than $M$ due to the unreliable network links. Furthermore, we only select a subset $\mathcal{I}_t$ of $\mathcal{I}_p$ for training in each round of FL in this paper, which can reduce both the computational and communication burden of participating clients. Similarly, we denote the number of selected clients $|\mathcal{I}_t|$ as $K$. Therefore, we have $\mathcal{I}_t \subset \mathcal{I}_p \subset \mathcal{I}$ and $K \leq M \leq N$. In this paper, we assume that each client $i, \forall i \in \mathcal{I}$ is associated with a local data source $Z_i$, where $Z_i = (X_i, Y_i)$ is a discrete random variable with the distribution function $P_{Z_i}$ and is with support on $\mathcal{Z}$. The sequence of random variables $Z_1, Z_2, ..., Z_N$ is denoted as $Z^{\mathcal{I}}$ and the corresponding joint distribution is denoted as $P_{Z^{\mathcal{I}}}$ in the following. The local distribution $P_{Z_i}$ of each data source $Z_i, \forall i \in \mathcal{I}$ is different from each other, i.e., $P_{Z_i} \neq P_{Z_j}, \forall i \neq j, i, j \in \mathcal{I}$, which is common in FL (Zhu et al., 2021).

The local training set $S_i = \{s_i^j\}_{j=1}^{n_i}, \forall i \in \mathcal{I}_p$ stored on participating client $i$ is made of $n_i$ i.i.d. realizations from the local data source $Z_i \sim P_{Z_i}$. Referring to Yuan et al. (2021), the objective of federated generalization is to train a global model on $\{S_i\}_{i \in \mathcal{I}_p}$, such that all the possible clients in $\mathcal{I}$ will be provided satisfactory service by this global model trained by participating clients. Let $\mathcal{H}$ be a hypothesis class on $\mathcal{X}$. The loss function $\ell : \mathcal{Y} \times \mathcal{Y} \to \mathbb{R}^+$ is a non-negative function. For simplicity, we denote $\ell(h(x), y)$ as $\ell(h, z)$ in the following. To understand the proposed framework better, we first present some definitions as follows:

**Definition 1** (Self-information weighted expected risk). The self-information of outcome $z_i \in \mathcal{Z}$ is denoted by $\log(\frac{1}{P(z_i)})$. Then, the self-information weighted expected risk is defined by

$$\mathcal{L}_{Z_i}(h) := \mathbb{E}\Big[\ell(h, Z_i) \log(\frac{1}{P(Z_i)})\Big] = \sum_{z_i \in \mathcal{Z}} P_{Z_i}(z_i)\ell(h, z_i) \log(\frac{1}{P_{Z_i}(z_i)}), \tag{1}$$

where $h$ is a specific model in $\mathcal{H}$ and $\ell(h, z_i)$ is the loss function of model $h$ on sample $z_i$.

The rationale behind the formulation of the risk outlined in equation 1 can be attributed to the fact that target distributions are unknown in the training stage under the OOD setting. It is reasonable to assume that unknown data sources have maximum entropy distributions, signifying greater uncertainty, exemplified by a uniform distribution of discrete labels. For example, IoT devices may only collect data in specific areas, while the global model trained by these devices is expected to provide spatial-related services for all devices across the entire area. Hence, the self-information of each outcome is

indispensable for measuring the expected risk in such a situation. Additionally, the proposed loss $\ell(h, z_i) \log(\frac{1}{P_{Z_i}(z_i)})$, is indicative of the requisite focus on outcomes $z_i \in \mathcal{Z}$ with lower probabilities in source distributions since they may have higher probabilities in unknown target distributions.

**Definition 2** (Joint self-information weighted expected risk). The self-information of one combination of outcomes $z_1, z_2, ..., z_N$ is denoted by $\log(\frac{1}{P_{Z^{\mathcal{I}}}(z_1,...,z_N)})$. We use the term $\frac{1}{N} \sum_{i \in \mathcal{I}} \ell(h, z_i)$ as the loss in this risk. This term denotes the average loss of model $h$ predicated on $z_1, z_2, ..., z_N$ and it is commonly used in the field of OOD generalization (Christiansen et al., 2022). Then, the joint self-information weighted expected risk on multiple data sources is defined by

$$
\begin{aligned}
\mathcal{L}_{Z^{\mathcal{I}}}(h) &:= \mathbb{E}_{Z^{\mathcal{I}}} \Big[ \frac{1}{N} \sum_{i \in \mathcal{I}} \ell(h, z_i) \log(\frac{1}{P(Z^{\mathcal{I}})}) \Big] \\
&= \sum_{z_1 \in \mathcal{Z}} \sum_{z_2 \in \mathcal{Z}} ... \sum_{z_N \in \mathcal{Z}} P_{Z^{\mathcal{I}}}(z_1, z_2, ..., z_N) \frac{1}{N} \sum_{i \in \mathcal{I}} \ell(h, z_i) \log(\frac{1}{P_{Z^{\mathcal{I}}}(z_1, z_2, ..., z_N)}),
\end{aligned}
\tag{2}
$$

where $h$ is a specific model in hypothesis space $\mathcal{H}$. Similarly, the self-information $\log(\frac{1}{P_{Z^{\mathcal{I}}}(z_1,...,z_N)})$ reflects the uncertainty of the event that outcomes $z_1, z_2, ..., z_N$ are sampled from $Z_1, Z_2, ..., Z_N$ respectively. We also denote $\mathcal{L}_{Z^{\mathcal{I}}}(h)$ as $\sum_{\mathcal{Z}^{\mathcal{I}}} P_{Z^{\mathcal{I}}} \frac{1}{N} \sum_{i \in \mathcal{I}} \ell(h, z_i) \log(\frac{1}{P_{Z^{\mathcal{I}}}})$ in the following.

Analogously, the motivation for defining the risk in equation 2 is that distributions of unparticipating clients may differ from distributions of participating clients significantly. We must account for the self-information of every possible combination of outcomes in order to ensure that one model performing well on participating clients will also do well on unparticipating clients with distinct distributions. Moving forward, we introduce the general training objective in FL, as well as the self-information weighted semi-empirical risk which we propose in this paper, respectively. The empirical risk minimization (ERM) objective in federated learning (Hu et al., 2023) is formulated as follows: $\mathcal{L}_S(h) := \sum_{i \in \mathcal{I}_p} \frac{\alpha_i}{n_i} \sum_{j=1}^{n_i} \ell(h, s_i^j), \alpha_i \geq 0, \sum_{i \in \mathcal{I}_p} \alpha_i = 1$, where $s_i^j$ denotes the $j$-th training sample at $i$-th selected client and $\alpha_i$ is the weighting factor of client $i$.

Let $S_i = \{s_i^j\}_{j=1}^{n_i}$ be the local training set of $i$-th participating client based on i.i.d. realizations from examples of the data source $Z_i$. $S$ is the whole training set over all the participating clients defined as $S = \cup_{i \in \mathcal{I}_p} S_i$. The empirical risk minimizer $\hat{h}$ is denoted as $\hat{h} = \arg\inf_{h \in \mathcal{H}} \mathcal{L}_S(h)$. Motivated by the definition of semi-empirical risk $\frac{1}{|\mathcal{I}|} \sum_{i \in \mathcal{I}} \mathbb{E}_{Z \sim P_{Z_i}}[\ell(h, Z)]$ in Hu et al. (2023), we further propose the self-information weighted semi-empirical risk as follows. The self-information weighted semi-empirical risk $\mathcal{L}_{Z_{\mathcal{I}_p}}(h)$ rooted on data source $\{Z_i\}_{i \in \mathcal{I}_p}$ is defined by $\mathcal{L}_{Z_{\mathcal{I}_p}}(h) := \sum_{i \in \mathcal{I}_p} \alpha_i \mathbb{E} \Big[ \ell(h, Z_i) \log(\frac{1}{P(Z_i)}) \Big]$. The semi-empirical risk minimizer $\hat{h}^*$ is denoted as $\hat{h}^* = \arg\inf_{h \in \mathcal{H}} \mathcal{L}_{Z_{\mathcal{I}_p}}(h)$. For participating clients $\mathcal{I}_p$, the corresponding semi-excess risk is defined as $\mathcal{L}_{Z_{\mathcal{I}_p}}(\hat{h}) - \mathcal{L}_{Z_{\mathcal{I}_p}}(\hat{h}^*)$ (Hu et al., 2023). The semi-excess risk represents how well the $\hat{h}$ can perform on the unseen data from $\{Z_i\}_{i \in \mathcal{I}_p}$ from the view of information theory.

### 3.2 FEDERATED GENERALIZATION

We now formally present the introduction of our proposed information-theoretic generalization framework in FL. We first define the information-theoretic generalization gap in FL as follows,

**Definition 3** (Information theoretic-generalization gap in federated learning).

$$
|\mathcal{L}_{Z^{\mathcal{I}}}(\hat{h}^*) - \mathcal{L}_{Z_{\mathcal{I}_p}}(\hat{h}^*)| := \Big| \mathbb{E}_{Z^{\mathcal{I}}} \Big[ \frac{1}{N} \sum_{i \in \mathcal{I}} \ell(\hat{h}^*, z_i) \log(\frac{1}{P(Z^{\mathcal{I}})}) \Big] - \sum_{i \in \mathcal{I}_p} \alpha_i \mathbb{E} \Big[ \ell(\hat{h}^*, Z_i) \log(\frac{1}{P(Z_i)}) \Big] \Big|,
\tag{3}
$$

where $\alpha_i \geq 0, \sum_{i \in \mathcal{I}_p} \alpha_i = 1$.

The motivation for defining the gap in equation 3 is that we want to know whether the global model $\hat{h}^*$ achieving the best on data sources of participating clients $\mathcal{I}_p$ via FL can also perform well on data sources of all the possible clients $\mathcal{I}$. We will find that the upper bound of this generalization gap $\mathcal{L}_{Z^{\mathcal{I}}}(\hat{h}^*) - \mathcal{L}_{Z_{\mathcal{I}_p}}(\hat{h}^*)$ includes the semi-excess risk $\mathcal{L}_{Z_{\mathcal{I}_p}}(\hat{h}) - \mathcal{L}_{Z_{\mathcal{I}_p}}(\hat{h}^*)$ defined above.

Furthermore, we decompose the original generalization gap in equation 3 and amplify it as follows,

$$\left|\mathcal{L}_{Z^{\mathcal{I}}}(\hat{h}^*) - \mathcal{L}_{Z_{\mathcal{I}_p}}(\hat{h}^*)\right| \leq \underbrace{\sup_{h \in \mathcal{H}} \left|\mathcal{L}_{Z^{\mathcal{I}}}(\hat{h}^*) - \mathcal{L}_{Z^{\mathcal{I}}}(h)\right|}_{\text{overfitting error}} + \underbrace{\left|\mathcal{L}_{Z^{\mathcal{I}}}(\hat{h}) - \mathcal{L}_{Z_{\mathcal{I}_p}}(\hat{h})\right|}_{\text{participation gap}} + \underbrace{\mathcal{L}_{Z_{\mathcal{I}_p}}(\hat{h}) - \mathcal{L}_{Z_{\mathcal{I}_p}}(\hat{h}^*)}_{\text{semi-excess risk}},$$

$$(4)$$

We conduct our theoretical analysis using the following assumptions.

**Assumption 1** (Bounded and Lipschitz loss). The loss function $\ell(h, z)$ is bounded by $b$. Besides, the loss function is differentiable with respect to $h$ and $L$-Lipschitz for every $z$, i.e., $\forall h, g \in \mathcal{H}, |\ell(h, z) - \ell(g, z)| \leq L\|h - g\|$. This assumption is commonly used in most of studies on generalization analysis (Deng et al., 2023; Sun et al., 2021).

**Assumption 2** (Limited Independence). The participating data sources $Z^{\mathcal{I}_p}$ are independent of the unparticipating data sources $Z^{\mathcal{I} \setminus \mathcal{I}_p}$, i.e., $P_{Z^{\mathcal{I}}} = P_{Z^{\mathcal{I}_p}} P_{Z^{\mathcal{I} \setminus \mathcal{I}_p}}$. In addition, among the participating data sources, each single data source $Z_i, \forall i \in \mathcal{I}_p$ is independent of the sequence of other participating data sources $Z^{\mathcal{I}_p \setminus i}$, i.e., $P_{Z^{\mathcal{I}_p}} = P_{Z_i} P_{Z^{\mathcal{I}_p \setminus i}}, \forall i \in \mathcal{I}_p$. This assumption is also reasonable since clients construct their own data sources spontaneously in reality.

Rooted on the above assumptions, we can derive three lemmas introduced in the appendix and we can further have below theorem about the information-theoretic generalization gap in FL.

**Theorem 1** (Information entropy-aware generalization gap in FL). *Let $\mathcal{G}$ be a family of functions related to hypothesis space $\mathcal{H} : z \mapsto \ell(h, z) : h \in \mathcal{H}$ with VC dimension $VC(\mathcal{G})$. For any $\delta \geq 0$, it follows that with probability at least $1 - \delta$,*

$$|\mathcal{L}_{Z^{\mathcal{I}}}(\hat{h}^*) - \mathcal{L}_{Z_{\mathcal{I}_p}}(\hat{h}^*)| \leq \underbrace{L\|\hat{h}^* - h^*\|H(Z^{\mathcal{I}})}_{\text{overfitting error}} + \underbrace{3bH(Z^{\mathcal{I}}) - b\sum_{i \in \mathcal{I}_p} \alpha_i H(Z_i)}_{\text{participation gap}}$$

$$\underbrace{+ \mathcal{E}_p + cb\sqrt{\frac{VC(\mathcal{G})}{\sum_{i=1}^{|\mathcal{I}_p|} n_i}} + b\sqrt{\frac{\log(1/\delta)}{2\sum_{i=1}^{|\mathcal{I}_p|} n_i}}}_{\text{semi-excess risk}},$$

$$(5)$$

*where $c$ is a constant. $h^* := \sup_{h \in \mathcal{H}} L\|\hat{h}^* - h\|H(Z^{\mathcal{I}})$ and $L$ is the Lipschitz constant. $\mathcal{E}_p = 2b\sum_{i \in \mathcal{I}_p} \sum_{z_i \in \mathcal{Z}} P_{Z_i}^2$.*

*Remark* 1. The first term $L\|\hat{h}^* - h^*\|H(Z^{\mathcal{I}})$ in equation 5 includes the distance $\|\hat{h}^* - h^*\|$ between the optimal model $\hat{h}^*$ trained on $Z^{\mathcal{I}_p}$ via FL and other models $h^*$. This gap $L\|\hat{h}^* - h^*\|H(Z^{\mathcal{I}})$ is affected more by greater $H(Z^{\mathcal{I}})$. The second term denotes that both the greater $\sum_{i \in \mathcal{I}_p} \alpha_i H(Z_i)$ and the smaller $H(Z^{\mathcal{I}})$ will decrease this generalization gap, which implies that the trained model will be powerful on unknown test domains if the model has encountered a sufficient number of outcomes during the training stage. The last term is the IID semi-excess risk for participating clients including the model and sample complexity.

In the following, we consider only leveraging a selected subset $\mathcal{I}_t \subset \mathcal{I}_p$ with the identical weighting factor $\alpha_i = \alpha_j = \frac{1}{|\mathcal{I}_t|} = \frac{1}{K}, \forall i \neq j, i, j \in \mathcal{I}_t$ for each selected client to measure the generalization gap in FL. In other words, we turn to focus on the below objective,

$$|\mathcal{L}_{Z^{\mathcal{I}}}(\hat{h}_t^*) - \mathcal{L}_{Z_{\mathcal{I}_t}}(\hat{h}_t^*)| = \left|\mathbb{E}_{Z^{\mathcal{I}}}\left[\frac{1}{N}\sum_{i \in \mathcal{I}} \ell(\hat{h}_t^*, Z_i) \log(\frac{1}{P(Z^{\mathcal{I}})})\right] - \sum_{i \in \mathcal{I}_t} \frac{1}{K}\mathbb{E}\left[\ell(\hat{h}_t^*, Z_i) \log(\frac{1}{P(Z_i)})\right]\right|$$

$$(6)$$

where $\hat{h}_t^* = \arg\inf_{h \in \mathcal{H}} \mathcal{L}_{Z_{\mathcal{I}_t}}(h)$.

Referring to the derivation presented in the proof of Theorem 1 in the appendix, we can further establish another theorem that pertains to the information-theoretical generalization gap in FL.

**Theorem 2** (Distribution discrepancy-aware generalization gap in FL). *Let $\mathcal{G}$ be a family of functions related to hypothesis space $\mathcal{H} : z \mapsto \ell(h, z) : h \in \mathcal{H}$ with VC dimension $VC(\mathcal{G})$. For any $\delta \geq 0$, it*

*follows that with probability at least $1 - \delta$,*

$$|\mathcal{L}_{Z^\mathcal{I}}(\hat{h}_t^*) - \mathcal{L}_{Z_{\mathcal{I}_t}}(\hat{h}_t^*)| \le L\|\hat{h}_t^* - h_t^*\|H(Z^\mathcal{I}) + b(3 - \frac{2M}{N})H(Z^\mathcal{I}) + b(2 - \frac{2K}{M} - \frac{1}{K})H(Z^{\mathcal{I}_p})$$

$$+ \frac{b}{K}\sum_{i \in \mathcal{I}_p \setminus \mathcal{I}_t} H(P_{Z_i}, P_{Z_j}) + \mathcal{E}_t + cb\sqrt{\frac{VC(\mathcal{G})}{\sum_{i=1}^{|\mathcal{I}_t|} n_i}} + b\sqrt{\frac{\log(1/\delta)}{2\sum_{i=1}^{|\mathcal{I}_t|} n_i}}, \forall j \ne i, j \in \mathcal{I}_p$$

(7)

*where $c$ is a constant. $h_t^* := \sup_{h \in \mathcal{H}} L\|\hat{h}_t^* - h\|H(Z^\mathcal{I})$ and $L$ is the Lipschitz constant. $\mathcal{E}_t = 2b\sum_{i \in \mathcal{I}_t}\sum_{z_i \in \mathcal{Z}} P_{Z_i}^2$. $H(P_{Z_i}, P_{Z_j}) = \mathbb{E}_{P_{Z_i}}[\log \frac{1}{P_{Z_j}}]$ is the cross entropy between distributions $P_{Z_i}$ and $P_{Z_j}$, measuring the dissimilarity between the two distributions.*

*Remark* 2. Theorem 2 indicates that lower dissimilarity $H(P_{Z_i}, P_{Z_j})$ between unselected distributions $P_{Z_i}, i \in \mathcal{I}_p \setminus \mathcal{I}_t$ and other participating distributions $P_{Z_j}, j \in \mathcal{I}_p$ can reduce the generalization gap. In other words, Theorem 2 reveals that participating data sources contains redundant information and a subset of $\mathcal{I}_p$ is adequate to represent the entirety. Notice that we do not need to compute the cross entropy $H(P_{Z_i}, P_{Z_j})$ in practice, the derived bound only inspire us to design the client selection algorithms to improve the generalization of FL.

## 4 METHODS

This section focuses on introducing a weighting aggregation approach, along with two client selection methods in FL, that are rooted on the theoretical findings mentioned earlier. The objective of employing these methods is to enhance the generalization performance of FL. Further elaboration on the procedure of the proposed methods can be found in the appendix, providing additional details.

### 4.1 MAXIMUM ENTROPY AGGREGATION

Inspired by Theorem 1, it is easy to find that to minimize the information-theoretic generalization gap is actually to maximize the term as follows

$$\max_{\{\alpha_i\}_{i \in \mathcal{I}_p}} \sum_{i \in \mathcal{I}_p} \alpha_i H(Z_i),$$

(8)

where $\sum_{i \in \mathcal{I}_p} \alpha_i = 1, \alpha_i \ge 0$, $H(Z_i)$ is the information entropy of data source $Z_i$.

The true entropy of each client in equation 8 is inaccessible in practice, so we can not assign $\alpha_i = 1$ for $i = \arg\max_{\tilde{i} \in \mathcal{I}_p} H(Z_{\tilde{i}})$ directly. Therefore, we can design proper weighting factors of local gradients in federated aggregation to maximize this term.

**Empirical entropy-based weighting:** Based on the above analysis, the weighting factor of local gradient $\nabla F_i(\mathbf{w}), i \in \mathcal{I}_p$ should be increased proportionally to the information entropy $H(Z_i)$ of the data source $Z_i$. This paper considers only the label distribution skew scenario for verifying the proposed empirical entropy-based weighting method. Therefore, we can design the aggregation weighting factor as follows:

$$\alpha_i = \frac{\exp(\hat{H}_i)}{\sum_{i \in \mathcal{I}_p} \exp(\hat{H}_i)}.$$

(9)

In this paper, we only consider the label skew scenario of heterogeneous FL, the empirical entropy can be thus calculated via $\hat{H}_i = -\sum_{y \in \mathcal{Y}} \frac{\sum_j \mathbb{I}_{y = y_j^i}}{n_i} \log \frac{\sum_j \mathbb{I}_{y = y_j^i}}{n_i}$, where $y_j^i$ is the label of $j$-th sample of local dataset $S_i$ of client $i$ and $\mathbb{I}$ denotes the indicator function. In fact, the proposed maximum entropy aggregation can be applied into other distribution shift scenarios directly if we can estimate the empirical entropy of data source (Paninski, 2003). How to leverage this aggregation method to benefit the federated generalization for other scenarios is out of the scope of this paper.

### 4.2 GRADIENT SIMILARITY-BASED CLIENT SELECTION

**Assumption 3** (Bounded dissimilarity). The dissimilarity between two local gradients is bounded by the divergence of corresponding local data distributions, i.e,

$$\|\nabla F_i(\mathbf{w}) - \nabla F_j(\mathbf{w})\|^2 \le \sigma_\mathcal{K}^2, \forall i, j \in \mathcal{K},$$

(10)

where $\nabla F_i(\mathbf{w})$ denotes the local gradient of client $i$ and $\sigma_{\mathcal{K}}^2$ depends on the dissimilarity of data distributions $\{P_{Z_k}\}_{k \in \mathcal{K}}$. In other words, a higher level of distribution dissimilarity of $\{P_{Z_k}\}_{k \in \mathcal{K}}$ will incur a higher level of $\sigma_{\mathcal{K}}^2$ (Zou et al., 2023; Li et al., 2020). Hence, $\sigma_{\mathcal{K}}^2$ is related to the term $\sup_{i,j \in \mathcal{K}} KL(P_{Z_i} \| P_{Z_j})$ in fact. Assumption 3 and Theorem 2 can help us to design client selection methods for improving the generalization performance of FL in the following.

We begin by introducing the general procedure of client selection methods. The server basically repeats the following steps in each round of FL: **a) Update the gradient table**: After updating the global model based on gradients uploaded by clients $i, \forall i \in \mathcal{I}_t$ following $\mathbf{w}_{t+1} \leftarrow \mathbf{w}_t - \sum_{i \in \mathcal{I}_t} \alpha_i \mathbf{g}_t^i$, the server updates and maintains a table to store the latest gradients uploaded by clients selected in each round. Specifically, the server performs the below actions: $\mathbf{g}_s^i \leftarrow \mathbf{g}_t^i, \forall i \in \mathcal{I}_t$ and $\mathbf{g}_s^i \leftarrow \mathbf{g}_s^i, \forall i \in \mathcal{I}_p \setminus \mathcal{I}_t$ to maintain the table $\{\mathbf{g}_s^i\}_{i \in \mathcal{I}_p}$. **b) Execute the selection algorithm**: Based on the gradients stored in the table $\{\mathbf{g}_s^i\}_{i \in \mathcal{I}_p}$, the server executes the client selection algorithms introduced below to determine the subset $\mathcal{I}_{t+1}$ of $\mathcal{I}_p$ to participate in the next round of FL. Note that this procedure eliminates the need for all participating clients to upload their gradients to the server in each round.

### 4.2.1 MINIMAX GRADIENT SIMILARITY-BASED CLIENT SELECTION

Recall the Theorem 2, to minimize the distribution discrepancy-aware generalization gap is in fact to maximize the term $\sum_{i \in \mathcal{I}_t} \min_{j \in \mathcal{I}_p} H(P_{Z_i}, P_{Z_j})$, which means that selecting clients with distributions $P_{Z_i}, i \in \mathcal{I}_t$ that differ significantly from other participating distributions $P_{Z_j}, \forall j \in \mathcal{I}_p$ can improve the generalization performance of FL. Furthermore, in reference to the Assumption 3, this objective can be formulated as a tractable optimization problem, given by

$$\min_{\{a_t^i\}_{i \in \mathcal{I}_p}} \sum_{i \in \mathcal{I}_p} a_t^i \max_{j \in \mathcal{I}_p, j \neq i} S(\nabla F_i(\mathbf{w}), \nabla F_j(\mathbf{w})), \tag{11}$$

$$s.t. \quad a_t^i \in \{0, 1\}, \forall i \in \mathcal{I}_p; \sum_{i \in \mathcal{I}_p} a_t^i = |\mathcal{I}_t|. \tag{11a}$$

where $S(\nabla F_i(\mathbf{w}), \nabla F_j(\mathbf{w}))$ represents the similarity metric between $\nabla F_i(\mathbf{w})$ and $\nabla F_j(\mathbf{w})$ uploaded by client $i$ and client $j$ respectively, is evaluated using cosine similarity in this work. This choice is motivated by the fact that the distance between bounded local gradients can be bounded by the distribution discrepancy between the corresponding local distributions, i.e, $\|\nabla_{\mathbf{w}} \mathcal{L}_{P_{Z_i}}(z; \mathbf{w}) - \nabla_w \mathcal{L}_{P_{Z_j}}(z; \mathbf{w})\|^2 \leq c' KL(P_{Z_i} \| P_{Z_j}) \leq c' H(P_{Z_i}, P_{Z_j})$, where $c'$ is a constant. Besides, the cosine similarity is related to the Euclidean distance: $\|\nabla F_i - \nabla F_j\|^2 = \|\nabla F_i\|^2 - 2 < \nabla F_i, \nabla F_j > + \|\nabla F_j\|^2$ and it provides a more stable approach in FL (Zeng et al., 2023). We propose a feasible approximate method for solving this optimization problem in Algorithm 1. Additionally, in Algorithm 1, we define the concept of "similarity set" denoted as $\mathbb{S}_i, \forall i \in \mathcal{I}_p$, which is a set of gradient similarities between the local gradient of each participating client and the gradients calculated by the other participating clients.

### 4.2.2 CONVEX HULL CONSTRUCTION-BASED CLIENT SELECTION

Besides, we can also minimize the generalization gap in Theorem 2 via maximizing a term as follows,

$$\max_{\mathcal{I}_t} \sum_{i \in \mathcal{I}_t} \sum_{j \in \mathcal{I}_p} H(P_{Z_i}, P_{Z_j}). \tag{12}$$

The above objective suggests that the larger difference between distributions $P_{Z_i}, \forall i \in \mathcal{I}_t$ and other participating distributions $P_{Z_j}, \forall j \in \mathcal{I}_p$, the smaller generalization gap in Theorem 2. Following the insights gained from our analysis and Assumption 3, a convex hull construction-based client selection policy is proposed to enhance the generalization of FL. The key idea of such method is to identify the vertices of the convex hull of gradients, and then select the corresponding clients whose gradients are located on the vertices of the constructed convex hull.

## 5 EXPERIMENT

In this section, we evaluate the proposed methods on three common datasets in FL, in order to verify our theoretical results. More details of experimental settings and results are provided in the appendix.

---

**Algorithm 1:** Minimax Gradient Similarity-based Client Selection Policy

---

    **Input:** set of local gradients $\{\mathbf{g}_t^i\}_{i \in \mathcal{I}_p}$, the size of selected clients $|\mathcal{I}_t|$ in round $t$.

1   Initialize collection of similarity sets $\{\mathbb{S}_i\}_{i \in \mathcal{I}_p}$, where $\mathbb{S}_i = \emptyset, \forall i \in \mathcal{I}_p$. Max similarity set
     $\mathbb{S}_{max} = \emptyset$. Candidate set $\tilde{\mathcal{I}}_t = \emptyset$ ;

2   **for** $i \in \mathcal{I}_p$ **do**

3      **for** $j \in \mathcal{I}_p, j \neq i$ **do**

4          $S_{i,j} \leftarrow S(\mathbf{g}_t^i, \mathbf{g}_t^j), \mathbb{S}_i \leftarrow \mathbb{S}_i \cup \{S_{i,j}\}$

5      **end**

6      $\mathbb{S}_{max} \leftarrow \mathbb{S}_{max} \cup \left\{S_i^* = \max_{S_{i,j}} \mathbb{S}_i\right\}$

7   **end**

8   **while** $|\tilde{\mathcal{I}}_t| \leq |\mathcal{I}_t|$ **do**

9      $\tilde{\mathcal{I}}_t \leftarrow \tilde{\mathcal{I}}_t \cup \left\{\arg\min_i \mathbb{S}_{max}\right\}, S^* \leftarrow \min_{S_i^*} \mathbb{S}_{max}, \mathbb{S}_{max} \leftarrow \mathbb{S}_{max} \setminus \{S^*\};$

10   **end**

11   **return** $\tilde{\mathcal{I}}_t$

---

---

**Algorithm 2:** Federated Generalization via Client Selection

---

    **Input:** set of local datasets $\{S_i\}_{i \in \mathcal{I}_p}$, participating client set $\mathcal{I}_p$ with size $|\mathcal{I}_p|$, the size of
     selected client set $|\mathcal{I}_t|$, the initial model $\mathbf{w}_0$, the local learning rate $\eta$.

1   Before the beginning of FL, all the participating clients update local models via $\mathbf{w}_0$ and
     upload local gradients $\mathbf{g}_0^i$ to the server;

2   **for** *round* $t \in [T]$ **do**

3      $\mathcal{I}_t \leftarrow$ Convex Hull($\{\mathbf{g}_s^i\}_{i \in \mathcal{I}_p}$) or $\mathcal{I}_t \leftarrow$ MinimaxSim($\{\mathbf{g}_s^i\}_{i \in \mathcal{I}_p}, |\mathcal{I}_t| = K$) ;

4      **for** *client* $i \in \mathcal{I}_t$ *in parallel* **do**

5          $\mathbf{w}_t^i \leftarrow LocalSolver(\mathbf{w}_t, S_i, \eta), \mathbf{g}_t^i \leftarrow \mathbf{w}_t - \mathbf{w}_t^i;$

6      **end**

7      $\mathbf{w}_{t+1} \leftarrow \mathbf{w}_t - \sum_{i \in \mathcal{I}_t} \alpha_i \mathbf{g}_t^i;$

8   **end**

---

**Experiment setting:** We consider three datasets commonly used in FL: i) image classification on EMNIST-10 and CIFAR-10 with a CNN model, ii) next character prediction on Shakespeare with a RNN model. For the image classification task, we split the dataset into different clients using the Dirichlet distribution spitting method, and we compare the proposed weighting aggregation method and client selection methods with baselines provided below. For the Shakespeare task, each speaking role in each play is set as a local dataset (Caldas et al., 2018), and we only compare the proposed client selection methods with baselines on this dataset. We split all datasets into 100 clients, and randomly select 40 clients among them as participating clients. The remaining 60 clients are considered unparticipating clients. We evaluate the global model's performance on two metrics: In Distribution (ID) performance and Out-Of-Distribution (OOD) performance. The ID performance evaluates the global model on the local test set of selected clients, while the OOD performance evaluates the global model on a standard test set with a distribution equivalent to the total dataset of 100 clients. All the selected clients perform 5 local epochs before sending their updates. We use a batch size of 128 and tune the local learning rate $\eta$ over a $\{0.1, 0.01, 0.001\}$ grid in all experiments.

Then we introduce some baseline methods: a) **Random selection:** Participating clients are selected randomly with equal probability. b) **Maximum gradient similarity-based selection:** This baseline evaluates the proposed minimax gradient similarity-based client selection method. The server selects clients with the most similar local gradients. c) **Interior selection:** This baseline evaluates the proposed convex hull construction-based selection method. The server constructs the convex hull of local gradients and then selects clients with gradients in the interior randomly. d) **Full sampling:** All participating clients will be selected in each round. e) **Power-of-Choice selection:** The server selects clients with the largest loss values in the current round (Cho et al., 2020). f) **Data size-based**

**weighting:** The weighting factor of each client is proportional to the data size of its local dataset. g) **Equality weighting:** The weighting factor of each client in aggregation is set to $1/|\mathcal{I}_t|$.

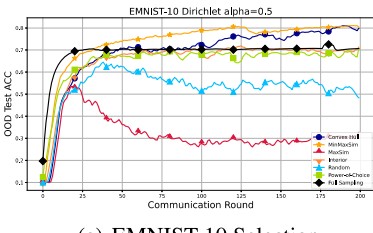
(a) EMNIST-10 Selection

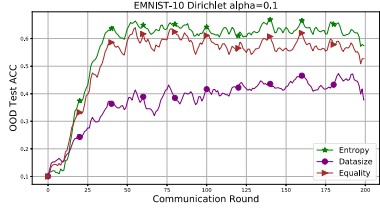
(b) EMNIST-10 Weighting

Figure 1: (a): OOD test accuracy of client selection methods for EMNIST-10; (b): OOD test accuracy of weighting methods for EMNIST-10.

**Experiment Results:** Table 1 reports the ID and OOD test accuracy of client selection methods on three datasets, while Figure 1 presents their convergence behaviors. The results in Table 1 indicate that, with respect to OOD test accuracy, the proposed client selection methods outperform random selection, Power-of-Choice selection, and other baselines. Furthermore, as shown in Figure 1, the proposed methods converge faster on OOD test accuracy than all the other baselines. These findings are consistent with our theoretical framework, which highlights that selecting clients with distributions distinct from other local distributions to participate in FL leads to improved generalization performance of the global model. Notably, on EMNIST-10 and CIFAR-10, the proposed methods perform even better than the full sampling baseline, possibly due to the randomness induced by client selection, which may enhance the generalization performance of FL. Table 2 demonstrates that the proposed empirical entropy-based weighting method surpasses the data size-based weighting and equality weighting method on OOD test accuracy and is more stable in convergence in Figure 1. This outcome aligns with our theoretical results and confirms that local models trained on distributions with greater information entropy contribute more significantly to federated generalization.

Table 1: Test accuracy (%) $\pm$ std of client selection methods

| Method | EMNIST-10 | | CIFAR-10 | | Shakespeare | |
|---|---|---|---|---|---|---|
| | ID | OOD | ID | OOD | ID | OOD |
| Convex Hull (ours) | 91.8±0.8 | **82.1±4.6** | 50.7±3.7 | **42.9±1.1** | 56.1±2.0 | **43.6±1.0** |
| MiniMaxSim (ours) | 95.5±0.7 | **82.3±4.7** | 49.9±3.0 | **42.0±1.0** | 55.0±1.9 | **43.5±0.8** |
| Random Selection | 95.9±0.9 | 69.4±5.4 | 49.9±2.6 | 38.9±1.0 | 45.2±0.6 | 37.3±1.1 |
| MaxSim | 95.3±0.5 | 59.8±6.3 | 60.9±2.1 | 32.2±3.1 | 30.2±0.8 | 22.2±1.4 |
| Interior | 96.6±0.5 | 73.8±6.5 | 50.4±5.7 | 40.8±0.9 | 33.5±3.6 | 25.4±0.8 |
| Full Sampling | 98.7±0.3 | 80.1±5.5 | 61.4±6.8 | 41.6±1.2 | 53.7±2.3 | 43.2±1.8 |
| Power-of-Choice | 97.3±0.6 | 76.3±4.5 | 60.3±6.7 | 39.2±1.3 | 56.7±2.6 | 42.6±1.7 |

Table 2: Test accuracy (%) $\pm$ std of weighting aggregation methods

| Method | EMNIST-10 | | CIFAR-10 | |
|---|---|---|---|---|
| | In Distribution | Out-Of-Distribution | In Distribution | Out-Of-Distribution |
| Entropy (ours) | 96.7±0.2 | **74.9±10.5** | 56.5±0.7 | **35.7±1.0** |
| Data size | 94.7±1.8 | 53.1±12.1 | 53.3±3.2 | 33.2±0.3 |
| Equality | 96.2±0.6 | 70.7±13.2 | 60.8±3.9 | 34.9±0.9 |

# 6 CONCLUSION

This paper addresses the generalization issue in FL by exploring whether a global model trained by participating clients is capable of performing well for unparticipating clients in the presence of heterogeneous data. To capture the generalization gap in FL, we propose an information-theoretic generalization framework that takes into account both the information entropy of local distribution and the discrepancy between different distributions. Leveraging this framework, we are able to identify the generalization gap and further propose an empirical entropy-based weighting aggregation method, as well as two gradient similarity-based client selection methods. These methods aim to improve the generalization performance of FL through distribution diversification. Numerical results corroborate our theoretical findings, demonstrating that the proposed approaches can surpass baselines.

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
