## A   PROOF OF THEOREMS IN FEDERATED GENERALIZATION

In this section, we will present detailed proofs of theorems in the theoretical framework section.

### A.1   PROOF OF THEOREM 1

*Proof.* Let us recall the decomposed generalization gap in FL,

$$
\begin{aligned}
\left| \mathcal{L}_{Z^{\mathcal{I}}}(\hat{h}^*) - \mathcal{L}_{Z_{\mathcal{I}_p}}(\hat{h}^*) \right| &= \left| \mathcal{L}_{Z^{\mathcal{I}}}(\hat{h}^*) - \mathcal{L}_{Z^{\mathcal{I}}}(\hat{h}) + \mathcal{L}_{Z^{\mathcal{I}}}(\hat{h}) - \mathcal{L}_{Z_{\mathcal{I}_p}}(\hat{h}) + \mathcal{L}_{Z_{\mathcal{I}_p}}(\hat{h}) - \mathcal{L}_{Z_{\mathcal{I}_p}}(\hat{h}^*) \right| \\
&\leq \left| \mathcal{L}_{Z^{\mathcal{I}}}(\hat{h}^*) - \mathcal{L}_{Z^{\mathcal{I}}}(\hat{h}) \right| + \left| \mathcal{L}_{Z^{\mathcal{I}}}(\hat{h}) - \mathcal{L}_{Z_{\mathcal{I}_p}}(\hat{h}) \right| + \left| \mathcal{L}_{Z_{\mathcal{I}_p}}(\hat{h}) - \mathcal{L}_{Z_{\mathcal{I}_p}}(\hat{h}^*) \right| \\
&\leq \underbrace{\sup_{h \in \mathcal{H}} \left| \mathcal{L}_{Z^{\mathcal{I}}}(\hat{h}^*) - \mathcal{L}_{Z^{\mathcal{I}}}(h) \right|}_{\text{overfitting error}} + \underbrace{\left| \mathcal{L}_{Z^{\mathcal{I}}}(\hat{h}) - \mathcal{L}_{Z_{\mathcal{I}_p}}(\hat{h}) \right|}_{\text{participation gap}} + \underbrace{\mathcal{L}_{Z_{\mathcal{I}_p}}(\hat{h}) - \mathcal{L}_{Z_{\mathcal{I}_p}}(\hat{h}^*)}_{\text{semi-excess risk}}.
\end{aligned}
$$

We regard the three terms in this decomposed generalization gap as three different lemmas and introduce the detailed proofs of these lemmas in the following.

**Lemma 1** (Overfitting error).

$$
\sup_{h \in \mathcal{H}} \left| \mathcal{L}_{Z^{\mathcal{I}}}(\hat{h}^*) - \mathcal{L}_{Z^{\mathcal{I}}}(h) \right| \leq L \| \hat{h}^* - h^* \| H(Z^{\mathcal{I}}),
$$

*where $h^* := \sup_{h \in \mathcal{H}} L \| \hat{h}^* - h \| H(Z^{\mathcal{I}})$, $L$ is the Lipschitz constant and $\| \hat{h}^* - h^* \|$ represents the distance between model $\hat{h}^*$ and model $h^*$.*

*Proof.*

$$
\begin{aligned}
\sup_{h \in \mathcal{H}} \left| \mathcal{L}_{Z^{\mathcal{I}}}(\hat{h}^*) - \mathcal{L}_{Z^{\mathcal{I}}}(h) \right| &= \sup_{h \in \mathcal{H}} \left| \sum_{\mathcal{Z}^{\mathcal{I}}} P_{Z^{\mathcal{I}}} \frac{1}{|\mathcal{I}|} \sum_{i \in \mathcal{I}} \log \left( \frac{1}{P_{Z^{\mathcal{I}}}} \right) [\ell(\hat{h}^*, z_i) - \ell(h, z_i)] \right| \\
&\leq \sup_{h \in \mathcal{H}} \left\{ \sum_{\mathcal{Z}^{\mathcal{I}}} P_{Z^{\mathcal{I}}} \log \left( \frac{1}{P_{Z^{\mathcal{I}}}} \right) \frac{1}{|\mathcal{I}|} \sum_{i \in \mathcal{I}} \left| \ell(\hat{h}^*, z_i) - \ell(h, z_i) \right| \right\} \\
&\leq \sup_{h \in \mathcal{H}} \left\{ \sum_{\mathcal{Z}^{\mathcal{I}}} P_{Z^{\mathcal{I}}} \log \left( \frac{1}{P_{Z^{\mathcal{I}}}} \right) \frac{1}{|\mathcal{I}|} \sum_{i \in \mathcal{I}} L \| \hat{h}^* - h \| \right\} \\
&= \sup_{h \in \mathcal{H}} L \| \hat{h}^* - h \| H(Z^{\mathcal{I}}) \\
&= L \| \hat{h}^* - h^* \| H(Z^{\mathcal{I}}),
\end{aligned}
$$

where $h^* := \sup_{h \in \mathcal{H}} L \| \hat{h}^* - h \| H(Z^{\mathcal{I}})$. The last inequality holds since the Assumption 1. $\qquad\square$

**Lemma 2** (Participation gap).

$$
\left| \mathcal{L}_{Z^{\mathcal{I}}}(\hat{h}) - \mathcal{L}_{Z_{\mathcal{I}_p}}(\hat{h}) \right| \leq 3bH(Z^{\mathcal{I}}) - b \sum_{i \in \mathcal{I}_p} \alpha_i H(Z_i).
$$

*Proof.*

$$\left| \mathcal{L}_{Z^{\mathcal{I}}}(\hat{h}) - \mathcal{L}_{Z^{\mathcal{I}_p}}(\hat{h}) \right| = \left| \sum_{\mathcal{Z}^{\mathcal{I}}} P_{Z^{\mathcal{I}}} \frac{1}{N} \sum_{i \in \mathcal{I}} \ell(\hat{h}, z_i) \log\left(\frac{1}{P_{Z^{\mathcal{I}}}}\right) - \sum_{i \in \mathcal{I}_p} \alpha_i \sum_{z_i \in \mathcal{Z}} P_{Z_i} \ell(\hat{h}, z_i) \log\left(\frac{1}{P_{Z_i}}\right) \right|$$

$$= \left| \sum_{\mathcal{Z}^{\mathcal{I}}} P_{Z^{\mathcal{I}}} \frac{1}{N} \sum_{i \in \mathcal{I}} \ell(\hat{h}, z_i) \log\left(\frac{1}{P_{Z^{\mathcal{I}}}}\right) - \sum_{\mathcal{Z}^{\mathcal{I}_p}} P_{Z^{\mathcal{I}_p}} \sum_{i \in \mathcal{I}_p} \alpha_i \ell(\hat{h}, z_i) \log\left(\frac{1}{P_{Z^{\mathcal{I}_p}}}\right) \right.$$

$$\left. + \sum_{\mathcal{Z}^{\mathcal{I}_p}} P_{Z^{\mathcal{I}_p}} \sum_{i \in \mathcal{I}_p} \alpha_i \ell(\hat{h}, z_i) \log\left(\frac{1}{P_{Z^{\mathcal{I}_p}}}\right) - \sum_{i \in \mathcal{I}_p} \alpha_i \sum_{z_i \in \mathcal{Z}} P_{Z_i} \ell(\hat{h}, z_i) \log\left(\frac{1}{P_{Z_i}}\right) \right|$$

$$\leq \underbrace{\left| \sum_{\mathcal{Z}^{\mathcal{I}}} P_{Z^{\mathcal{I}}} \frac{1}{N} \sum_{i \in \mathcal{I}} \ell(\hat{h}, z_i) \log\left(\frac{1}{P_{Z^{\mathcal{I}}}}\right) - \sum_{\mathcal{Z}^{\mathcal{I}_p}} P_{Z^{\mathcal{I}_p}} \sum_{i \in \mathcal{I}_p} \alpha_i \ell(\hat{h}, z_i) \log\left(\frac{1}{P_{Z^{\mathcal{I}_p}}}\right) \right|}_{\text{participation discrepancy}}$$

$$+ \underbrace{\left| \sum_{\mathcal{Z}^{\mathcal{I}_p}} P_{Z^{\mathcal{I}_p}} \sum_{i \in \mathcal{I}_p} \alpha_i \ell(\hat{h}, z_i) \log\left(\frac{1}{P_{Z^{\mathcal{I}_p}}}\right) - \sum_{i \in \mathcal{I}_p} \alpha_i \sum_{z_i \in \mathcal{Z}} P_{Z_i} \ell(\hat{h}, z_i) \log\left(\frac{1}{P_{Z_i}}\right) \right|}_{\text{distributed learning discrepancy}}.$$

Notice that the term called "participation discrepancy" mainly reflects the discrepancy between the whole data source $Z^{\mathcal{I}}$ and participating data source $Z^{\mathcal{I}_p}$. While the term called "distributed learning discrepancy" represents the gap between performing the centralized training on participating data source $Z^{\mathcal{I}_p}$ and conducting the distributed learning on the same source $Z^{\mathcal{I}_p}$. Now we first focus on the "participation discrepancy" term and we denote this term as $gen(Z^{\mathcal{I}}, Z^{\mathcal{I}_p}; \hat{h})$ in the following.

$$gen(Z^{\mathcal{I}}, Z^{\mathcal{I}_p}; \hat{h}) = \left| \sum_{\mathcal{Z}^{\mathcal{I}}} P_{Z^{\mathcal{I}}} \frac{1}{N} \sum_{i \in \mathcal{I}} \ell(\hat{h}, z_i) \log\left(\frac{1}{P_{Z^{\mathcal{I}}}}\right) - \sum_{\mathcal{Z}^{\mathcal{I}_p}} P_{Z^{\mathcal{I}_p}} \sum_{i \in \mathcal{I}_p} \alpha_i \ell(\hat{h}, z_i) \log\left(\frac{1}{P_{Z^{\mathcal{I}_p}}}\right) \right|$$

$$= \left| \sum_{\mathcal{Z}^{\mathcal{I}}} P_{Z^{\mathcal{I}}} \frac{1}{N} \sum_{i \in \mathcal{I}_p} \ell(\hat{h}, z_i) \log\left(\frac{1}{P_{Z^{\mathcal{I}}}}\right) + \sum_{\mathcal{Z}^{\mathcal{I}}} P_{Z^{\mathcal{I}}} \frac{1}{N} \sum_{i \in \mathcal{I} \setminus \mathcal{I}_p} \ell(\hat{h}, z_i) \log\left(\frac{1}{P_{Z^{\mathcal{I}}}}\right) \right.$$

$$\left. - \sum_{\mathcal{Z}^{\mathcal{I}_p}} P_{Z^{\mathcal{I}_p}} \sum_{i \in \mathcal{I}_p} \alpha_i \ell(\hat{h}, z_i) \log\left(\frac{1}{P_{Z^{\mathcal{I}_p}}}\right) \right|$$

$$\leq \left| \sum_{\mathcal{Z}^{\mathcal{I}}} P_{Z^{\mathcal{I}}} \frac{1}{N} \sum_{i \in \mathcal{I} \setminus \mathcal{I}_p} \ell(\hat{h}, z_i) \log\left(\frac{1}{P_{Z^{\mathcal{I}}}}\right) \right| + \left| \sum_{\mathcal{Z}^{\mathcal{I}}} P_{Z^{\mathcal{I}}} \frac{1}{N} \sum_{i \in \mathcal{I}_p} \ell(\hat{h}, z_i) \log\left(\frac{1}{P_{Z^{\mathcal{I}}}}\right) \right.$$

$$\left. - \sum_{\mathcal{Z}^{\mathcal{I}_p}} P_{Z^{\mathcal{I}_p}} \sum_{i \in \mathcal{I}_p} \alpha_i \ell(\hat{h}, z_i) \log\left(\frac{1}{P_{Z^{\mathcal{I}_p}}}\right) \right|$$

$$\leq \frac{1}{N} \sum_{i \in \mathcal{I} \setminus \mathcal{I}_p} b \left| \sum_{\mathcal{Z}^{\mathcal{I}}} P_{Z^{\mathcal{I}}} \log\left(\frac{1}{P_{Z^{\mathcal{I}}}}\right) \right| + \left| \sum_{\mathcal{Z}^{\mathcal{I}}} P_{Z^{\mathcal{I}}} \frac{1}{N} \sum_{i \in \mathcal{I}_p} \ell(\hat{h}, z_i) \log\left(\frac{1}{P_{Z^{\mathcal{I}}}}\right) \right.$$

$$\left. - \sum_{\mathcal{Z}^{\mathcal{I}_p}} P_{Z^{\mathcal{I}_p}} \sum_{i \in \mathcal{I}_p} \alpha_i \ell(\hat{h}, z_i) \log\left(\frac{1}{P_{Z^{\mathcal{I}_p}}}\right) \right|$$

$$= \frac{N-M}{N} b H(Z^{\mathcal{I}}) + \left| \sum_{\mathcal{Z}^{\mathcal{I}}} P_{Z^{\mathcal{I}}} \frac{1}{N} \sum_{i \in \mathcal{I}_p} \ell(\hat{h}, z_i) \log\left(\frac{1}{P_{Z^{\mathcal{I}}}}\right) - \sum_{\mathcal{Z}^{\mathcal{I}_p}} P_{Z^{\mathcal{I}_p}} \sum_{i \in \mathcal{I}_p} \alpha_i \ell(\hat{h}, z_i) \log\left(\frac{1}{P_{Z^{\mathcal{I}_p}}}\right) \right|$$

$$= \frac{N-M}{N} b H(Z^{\mathcal{I}}) + \left| \sum_{\mathcal{Z}^{\mathcal{I}}} P_{Z^{\mathcal{I}}} \frac{1}{N} \sum_{i \in \mathcal{I}_p} \ell(\hat{h}, z_i) \log\left(\frac{1}{P_{Z^{\mathcal{I}}}}\right) \right.$$

$$\left. - \sum_{\mathcal{Z}^{\mathcal{I}_p}} \sum_{\mathcal{Z}^{\mathcal{I} \setminus \mathcal{I}_p}} P_{Z^{\mathcal{I}_p}} P_{Z^{\mathcal{I} \setminus \mathcal{I}_p}} \sum_{i \in \mathcal{I}_p} \alpha_i \ell(\hat{h}, z_i) \log\left(\frac{1}{P_{Z^{\mathcal{I}_p}}}\right) \right|.$$

According to Assumption 2, we can have,

$$gen(Z^{\mathcal{I}}, Z^{\mathcal{I}_p}; \hat{h}) \leq \frac{N-M}{N} bH(Z^{\mathcal{I}}) + \Big| \sum_{\mathcal{Z}^{\mathcal{I}}} P_{Z^{\mathcal{I}}} \Big[ \frac{1}{N} \sum_{i \in \mathcal{I}_p} \ell(\hat{h}, z_i) \log\Big(\frac{1}{P_{Z^{\mathcal{I}}}}\Big) - \sum_{i \in \mathcal{I}_p} \alpha_i \ell(\hat{h}, z_i) \log\Big(\frac{1}{P_{Z^{\mathcal{I}_p}}}\Big) \Big] \Big|$$

$$\leq \frac{N-M}{N} bH(Z^{\mathcal{I}}) + \sum_{\mathcal{Z}^{\mathcal{I}}} P_{Z^{\mathcal{I}}} \sum_{i \in \mathcal{I}_p} \Big| \frac{1}{N} \ell(\hat{h}, z_i) \log\Big(\frac{1}{P_{Z^{\mathcal{I}}}}\Big) - \alpha_i \ell(\hat{h}, z_i) \log\Big(\frac{1}{P_{Z^{\mathcal{I}_p}}}\Big) \Big|$$

$$= \frac{N-M}{N} bH(Z^{\mathcal{I}}) + \sum_{\mathcal{Z}^{\mathcal{I}}} P_{Z^{\mathcal{I}}} \sum_{i \in \mathcal{I}_p} \Big| \frac{1}{N} \ell(\hat{h}, z_i) \log\Big(\frac{1}{P_{Z^{\mathcal{I}}}}\Big) - \alpha_i \ell(\hat{h}, z_i) \log\Big(\frac{1}{P_{Z^{\mathcal{I}}}}\Big)$$

$$+ \alpha_i \ell(\hat{h}, z_i) \log\Big(\frac{1}{P_{Z^{\mathcal{I}}}}\Big) - \alpha_i \ell(\hat{h}, z_i) \log\Big(\frac{1}{P_{Z^{\mathcal{I}_p}}}\Big) \Big|$$

$$\leq \frac{N-M}{N} bH(Z^{\mathcal{I}}) + \sum_{\mathcal{Z}^{\mathcal{I}}} P_{Z^{\mathcal{I}}} \sum_{i \in \mathcal{I}_p} \Big| \frac{1}{N} \ell(\hat{h}, z_i) \log\Big(\frac{1}{P_{Z^{\mathcal{I}}}}\Big) - \alpha_i \ell(\hat{h}, z_i) \log\Big(\frac{1}{P_{Z^{\mathcal{I}}}}\Big) \Big|$$

$$+ \sum_{\mathcal{Z}^{\mathcal{I}}} P_{Z^{\mathcal{I}}} \sum_{i \in \mathcal{I}_p} \Big| \alpha_i \ell(\hat{h}, z_i) \log\Big(\frac{1}{P_{Z^{\mathcal{I}}}}\Big) - \alpha_i \ell(\hat{h}, z_i) \log\Big(\frac{1}{P_{Z^{\mathcal{I}_p}}}\Big) \Big|$$

$$\leq \frac{N-M}{N} bH(Z^{\mathcal{I}}) + \sum_{\mathcal{Z}^{\mathcal{I}}} P_{Z^{\mathcal{I}}} \sum_{i \in \mathcal{I}_p} b \Big| \frac{1}{N} \log\Big(\frac{1}{P_{Z^{\mathcal{I}}}}\Big) - \alpha_i \log\Big(\frac{1}{P_{Z^{\mathcal{I}}}}\Big) \Big|$$

$$+ \sum_{\mathcal{Z}^{\mathcal{I}}} P_{Z^{\mathcal{I}}} \sum_{i \in \mathcal{I}_p} \alpha_i b \Big| \log\Big(\frac{1}{P_{Z^{\mathcal{I}}}}\Big) - \log\Big(\frac{1}{P_{Z^{\mathcal{I}_p}}}\Big) \Big|$$

$$\leq \frac{N-M}{N} bH(Z^{\mathcal{I}}) + \sum_{\mathcal{Z}^{\mathcal{I}}} P_{Z^{\mathcal{I}}} \sum_{i \in \mathcal{I}_p} b \Big[ \Big| \frac{1}{N} \log\Big(\frac{1}{P_{Z^{\mathcal{I}}}}\Big) \Big| + \Big| \alpha_i \log\Big(\frac{1}{P_{Z^{\mathcal{I}}}}\Big) \Big| \Big]$$

$$+ \sum_{\mathcal{Z}^{\mathcal{I}}} P_{Z^{\mathcal{I}}} \sum_{i \in \mathcal{I}_p} \alpha_i b \Big[ \log\Big(\frac{1}{P_{Z^{\mathcal{I}}}}\Big) - \log\Big(\frac{1}{P_{Z^{\mathcal{I}_p}}}\Big) \Big]$$

$$\leq \frac{N-M}{N} bH(Z^{\mathcal{I}}) + \sum_{\mathcal{Z}^{\mathcal{I}}} P_{Z^{\mathcal{I}}} \sum_{i \in \mathcal{I}_p} b(\frac{1}{N} + \alpha_i) \log\Big(\frac{1}{P_{Z^{\mathcal{I}}}}\Big)$$

$$+ \sum_{\mathcal{Z}^{\mathcal{I}}} P_{Z^{\mathcal{I}}} \sum_{i \in \mathcal{I}_p} \alpha_i b \Big[ \log\Big(\frac{1}{P_{Z^{\mathcal{I}}}}\Big) - \log\Big(\frac{1}{P_{Z^{\mathcal{I}_p}}}\Big) \Big]$$

$$= \frac{N-M}{N} bH(Z^{\mathcal{I}}) + b \sum_{\mathcal{Z}^{\mathcal{I}}} P_{Z^{\mathcal{I}}} (\frac{M}{N} + 1) \log\Big(\frac{1}{P_{Z^{\mathcal{I}}}}\Big) + b \sum_{i \in \mathcal{I}_p} \alpha_i \sum_{\mathcal{Z}^{\mathcal{I}}} P_{Z^{\mathcal{I}}} \log\Big(\frac{1}{P_{Z^{\mathcal{I}}}}\Big)$$

$$- b \sum_{i \in \mathcal{I}_p} \alpha_i \sum_{\mathcal{Z}^{\mathcal{I}_p}} P_{Z^{\mathcal{I}_p}} \log\Big(\frac{1}{P_{Z^{\mathcal{I}_p}}}\Big)$$

$$= \frac{N-M}{N} bH(Z^{\mathcal{I}}) + \frac{N+M}{N} bH(Z^{\mathcal{I}}) + bH(Z^{\mathcal{I}}) - bH(Z^{\mathcal{I}_p})$$

$$= 3bH(Z^{\mathcal{I}}) - bH(Z^{\mathcal{I}_p}).$$

Then we focus on the "distributed learning discrepancy" term and we denote it as $gen(Z^{\mathcal{I}_p}; \hat{h})$.

$$gen(Z^{\mathcal{I}_p}; \hat{h}) = \Big| \sum_{\mathcal{Z}^{\mathcal{I}_p}} P_{Z^{\mathcal{I}_p}} \sum_{i \in \mathcal{I}_p} \alpha_i \ell(\hat{h}, z_i) \log\Big(\frac{1}{P_{Z^{\mathcal{I}_p}}}\Big) - \sum_{i \in \mathcal{I}_p} \alpha_i \sum_{z_i \in \mathcal{Z}} P_{Z_i} \ell(\hat{h}, z_i) \log\Big(\frac{1}{P_{Z_i}}\Big) \Big|$$

$$\leq \sum_{i \in \mathcal{I}_p} \alpha_i \Big| \sum_{\mathcal{Z}^{\mathcal{I}_p}} P_{Z^{\mathcal{I}_p}} \ell(\hat{h}, z_i) \log\Big(\frac{1}{P_{Z^{\mathcal{I}_p}}}\Big) - \sum_{z_i \in \mathcal{Z}} P_{Z_i} \ell(\hat{h}, z_i) \log\Big(\frac{1}{P_{Z_i}}\Big) \Big|$$

$$= \sum_{i \in \mathcal{I}_p} \alpha_i \Big| \sum_{\mathcal{Z}^{\mathcal{I}_p}} P_{Z^{\mathcal{I}_p}} \ell(\hat{h}, z_i) \log\Big(\frac{1}{P_{Z^{\mathcal{I}_p}}}\Big) - \sum_{z_i \in \mathcal{Z}} \sum_{\mathcal{Z}^{\mathcal{I}_p \backslash i}} P_{Z_i} P_{Z^{\mathcal{I}_p \backslash i}} \ell(\hat{h}, z_i) \log\Big(\frac{1}{P_{Z_i}}\Big) \Big|.$$

Similarly, based on Assumption 2, we can also have,

$$gen(Z^{\mathcal{I}_p}; \hat{h}) \leq \sum_{i \in \mathcal{I}_p} \alpha_i \sum_{\mathcal{Z}^{\mathcal{I}_p}} P_{Z^{\mathcal{I}_p}} \Big| \ell(\hat{h}, z_i) \log \Big( \frac{1}{P_{Z^{\mathcal{I}_p}}} \Big) - \ell(\hat{h}, z_i) \log \Big( \frac{1}{P_{Z_i}} \Big) \Big|$$

$$\leq b \sum_{i \in \mathcal{I}_p} \alpha_i \sum_{\mathcal{Z}^{\mathcal{I}_p}} P_{Z^{\mathcal{I}_p}} \Big| \log \Big( \frac{1}{P_{Z^{\mathcal{I}_p}}} \Big) - \log \Big( \frac{1}{P_{Z_i}} \Big) \Big|$$

$$= b \sum_{i \in \mathcal{I}_p} \alpha_i \Big[ \sum_{\mathcal{Z}^{\mathcal{I}_p}} P_{Z^{\mathcal{I}_p}} \log \Big( \frac{1}{P_{Z^{\mathcal{I}_p}}} \Big) - \sum_{z_i \in \mathcal{Z}} P_{Z_i} \log \Big( \frac{1}{P_{Z_i}} \Big) \Big]$$

$$= bH(Z^{\mathcal{I}_p}) - b \sum_{i \in \mathcal{I}_p} \alpha_i H(Z_i).$$

To sum up, the participation gap can be bounded as follows,

$$\Big| \mathcal{L}_{Z^{\mathcal{I}}}(\hat{h}) - \mathcal{L}_{Z^{\mathcal{I}_p}}(\hat{h}) \Big| \leq 3bH(Z^{\mathcal{I}}) - b \sum_{i \in \mathcal{I}_p} \alpha_i H(Z_i).$$

$\square$

**Lemma 3** (Semi-excess risk). *Let $\mathcal{G}$ be a family of functions related to hypothesis space $\mathcal{H} : z \mapsto \ell(h, z) : h \in \mathcal{H}$ with VC dimension $VC(\mathcal{G})$. Distributed training sets $\{S_i\}_{i=1}^N = \{\{s_j^i\}_{j=1}^{n_i}\}_{i=1}^N$ are constructed by i.i.d. realizations sampled from different data sources $\{Z_i\}_{i=1}^N$. If the loss function $\ell$ is bounded by $b$, for any $\delta \geq 0$, it follows that with probability at least $1 - \delta$,*

$$\mathcal{L}_{Z_{\mathcal{I}_p}}(\hat{h}) - \mathcal{L}_{Z_{\mathcal{I}_p}}(\hat{h}^*) \leq \mathcal{E}_p + cb\sqrt{\frac{VC(\mathcal{G})}{\sum_{i=1}^{|\mathcal{I}_p|} n_i}} + b\sqrt{\frac{\log(1/\delta)}{2\sum_{i=1}^{|\mathcal{I}_p|} n_i}},$$

*where $c$ is a constant, and $\mathcal{E}_p = 2b \sum_{i \in \mathcal{I}_p} \sum_{z_i \in \mathcal{Z}} P_{Z_i}^2$. The term $\mathcal{E}_p$ represents the gap between the self-information weighted expected risk and the vanilla expected risk without considering the self-information of outcomes.*

Before stating the detailed proof of Lemma 3, we introduce a theoretical result about the generalization bound for participating clients in IID federated learning in (Hu et al., 2023) into our paper as a lemma.

**Lemma 4** (Generalization bound for participating clients in IID FL). *(Hu et al., 2023) Let $\mathcal{G}$ be a family of functions related to hypothesis space $\mathcal{H} : z \mapsto \ell(h, z) : h \in \mathcal{H}$ with VC dimension $VC(\mathcal{G})$. Distributed training sets $\{S_i\}_{i=1}^N = \{\{s_j^i\}_{j=1}^{n_i}\}_{i=1}^N$ are constructed by i.i.d. realizations sampled from different data sources $\{Z_i\}_{i=1}^N$. If the loss function $\ell$ is bounded by $b$, for any $\delta \geq 0$, it follows that with probability at least $1 - \delta$,*

$$\sup_{h \in \mathcal{H}} \Big| \sum_{i=1}^N \alpha_i \mathbb{E}(\ell(h, Z_i)) - \sum_{i=1}^N \frac{\alpha_i}{n_i} \sum_{j=1}^{n_i} \ell(h, s_j^i) \Big| \leq cb\sqrt{\frac{VC(\mathcal{G})}{\sum_{i=1}^N n_i}} + b\sqrt{\frac{ln(1/\delta)}{2\sum_{i=1}^N n_i}},$$

*where $c$ is a constant.*

And then we start the formal proof of the Lemma 3

*Proof.*

$$\mathcal{L}_{Z_{\mathcal{I}_p}}(\hat{h}) - \mathcal{L}_{Z_{\mathcal{I}_p}}(\hat{h}^*) = \underbrace{\mathcal{L}_{Z_{\mathcal{I}_p}}(\hat{h}) - \mathcal{L}_S(\hat{h})}_{\text{Term A}} + \underbrace{\mathcal{L}_S(\hat{h}) - \mathcal{L}_S(\hat{h}^*)}_{\text{ERM}} + \underbrace{\mathcal{L}_S(\hat{h}^*) - \mathcal{L}_{Z_{\mathcal{I}_p}}(\hat{h}^*)}_{\text{Term B}}.$$

The ERM term $\mathcal{L}_S(\hat{h}) - \mathcal{L}_S(\hat{h}^*) \leq 0$. Then we focus on the term A first.

$$\mathcal{L}_{Z_{\mathcal{I}_p}}(\hat{h}) - \mathcal{L}_S(\hat{h}) = \mathcal{L}_{Z_{\mathcal{I}_p}}(\hat{h}) - \sum_{i \in \mathcal{I}_p} \alpha_i \mathbb{E}[\ell(\hat{h}, z_i)] + \sum_{i \in \mathcal{I}_p} \alpha_i \mathbb{E}[\ell(\hat{h}, z_i)] - \mathcal{L}_S(\hat{h})$$

$$\leq \sup_{h \in \mathcal{H}} \left| \mathcal{L}_{Z_{\mathcal{I}_p}}(h) - \sum_{i \in \mathcal{I}_p} \alpha_i \mathbb{E}[\ell(h, z_i)] + \sum_{i \in \mathcal{I}_p} \alpha_i \mathbb{E}[\ell(h, z_i)] - \mathcal{L}_S(h) \right|$$

$$\leq \sup_{h \in \mathcal{H}} \left| \mathcal{L}_{Z_{\mathcal{I}_p}}(h) - \sum_{i \in \mathcal{I}_p} \alpha_i \mathbb{E}[\ell(h, z_i)] \right| + \sup_{h \in \mathcal{H}} \left| \sum_{i \in \mathcal{I}_p} \alpha_i \mathbb{E}[\ell(h, z_i)] - \mathcal{L}_S(h) \right|.$$

Rooted on the Lemma 4, we can immediately have the below result with probability at least $1 - \delta$,

$$\mathcal{L}_{Z_{\mathcal{I}_p}}(\hat{h}) - \mathcal{L}_S(\hat{h}) \leq \sup_{h \in \mathcal{H}} \left| \mathcal{L}_{Z_{\mathcal{I}_p}}(h) - \sum_{i \in \mathcal{I}_p} \alpha_i \mathbb{E}[\ell(h, z_i)] \right| + cb \sqrt{\frac{VC(\mathcal{G})}{\sum_{i=1}^{|\mathcal{I}_p|} n_i}} + b \sqrt{\frac{\log(1/\delta)}{2 \sum_{i=1}^{|\mathcal{I}_p|} n_i}}.$$

Now we focus on the term B, we can have:

$$\mathcal{L}_S(\hat{h}^*) - \mathcal{L}_{Z_{\mathcal{I}_p}}(\hat{h}^*) = \mathcal{L}_S(\hat{h}^*) - \sum_{i \in \mathcal{I}_p} \alpha_i \mathbb{E}[\ell(\hat{h}^*, z_i)] + \sum_{i \in \mathcal{I}_p} \alpha_i \mathbb{E}[\ell(\hat{h}^*, z_i)] - \mathcal{L}_{Z_{\mathcal{I}_p}}(\hat{h}^*).$$

The concentration term $\mathcal{L}_S(\hat{h}^*) - \sum_{i \in \mathcal{I}_p} \alpha_i \mathbb{E}[\ell(\hat{h}^*, z_i)]$ does not affect the bound because $\hat{h}^*$ is a specific model in hypothesis class. To sum up, we have:

$$\mathcal{L}_{Z_{\mathcal{I}_p}}(\hat{h}) - \mathcal{L}_{Z_{\mathcal{I}_p}}(\hat{h}^*) \leq \sup_{h \in \mathcal{H}} \left| \mathcal{L}_{Z_{\mathcal{I}_p}}(h) - \sum_{i \in \mathcal{I}_p} \alpha_i \mathbb{E}[\ell(h, z_i)] \right| + \sum_{i \in \mathcal{I}_p} \alpha_i \mathbb{E}[\ell(\hat{h}^*, z_i)] - \mathcal{L}_{Z_{\mathcal{I}_p}}(\hat{h}^*)$$

$$+ cb \sqrt{\frac{VC(\mathcal{G})}{\sum_{i=1}^{|\mathcal{I}_p|} n_i}} + b \sqrt{\frac{\log(1/\delta)}{2 \sum_{i=1}^{|\mathcal{I}_p|} n_i}}.$$

For simplicity, we denote the term $cb \sqrt{\frac{VC(\mathcal{G})}{\sum_{i=1}^{|\mathcal{I}_p|} n_i}} + b \sqrt{\frac{\log(1/\delta)}{2 \sum_{i=1}^{|\mathcal{I}_p|} n_i}}$ as $\mathcal{N}(\mathcal{G}, \mathcal{I}_p)$ temporarily.

Assume that $h' \in \mathcal{H}$ satisfying $\sup_{h \in \mathcal{H}} \left| \mathcal{L}_{Z_{\mathcal{I}_p}}(h) - \sum_{i \in \mathcal{I}_p} \alpha_i \mathbb{E}[\ell(h, z_i)] \right| = \sum_{i \in \mathcal{I}_p} \alpha_i \mathbb{E}[\ell(h', z_i)] - \mathcal{L}_{Z_{\mathcal{I}_p}}(h')$, we have

$$\mathcal{L}_{Z_{\mathcal{I}_p}}(\hat{h}) - \mathcal{L}_{Z_{\mathcal{I}_p}}(\hat{h}^*) \leq \sum_{i \in \mathcal{I}_p} \alpha_i \mathbb{E}[\ell(h', z_i)] - \mathcal{L}_{Z_{\mathcal{I}_p}}(h') + \sum_{i \in \mathcal{I}_p} \alpha_i \mathbb{E}[\ell(\hat{h}^*, z_i)] - \mathcal{L}_{Z_{\mathcal{I}_p}}(\hat{h}^*) + \mathcal{N}(\mathcal{G}, \mathcal{I}_p)$$

$$= \sum_{i \in \mathcal{I}_p} \alpha_i \mathbb{E}[\ell(\hat{h}^*, z_i) + \ell(h', z_i)] - \left( \mathcal{L}_{Z_{\mathcal{I}_p}}(h') + \mathcal{L}_{Z_{\mathcal{I}_p}}(\hat{h}^*) \right) + \mathcal{N}(\mathcal{G}, \mathcal{I}_p)$$

$$= \sum_{i \in \mathcal{I}_p} \alpha_i \sum_{z_i \in \mathcal{Z}} P_{Z_i} \left( \ell(h', z_i) + \ell(\hat{h}^*, z_i) \right)$$

$$- \sum_{i \in \mathcal{I}_p} \alpha_i \sum_{z_i \in \mathcal{Z}} P_{Z_i} \left( \ell(\hat{h}^*, z_i) + \ell(h', z_i) \right) \log \left( \frac{1}{P_{Z_i}} \right) + \mathcal{N}(\mathcal{G}, \mathcal{I}_p)$$

$$= \sum_{i \in \mathcal{I}_p} \alpha_i \sum_{z_i \in \mathcal{Z}} P_{Z_i} \left( \ell(\hat{h}^*, z_i) + \ell(h', z_i) \right) \left( \log(P_{Z_i}) + 1 \right) + \mathcal{N}(\mathcal{G}, \mathcal{I}_p)$$

$$\leq \sum_{i \in \mathcal{I}_p} \alpha_i \sum_{z_i \in \mathcal{Z}} P_{Z_i}^2 \left( \ell(\hat{h}^*, z_i) + \ell(h', z_i) \right) + \mathcal{N}(\mathcal{G}, \mathcal{I}_p)$$

$$\leq 2b \sum_{i \in \mathcal{I}_p} \sum_{z_i \in \mathcal{Z}} P_{Z_i}^2 + \mathcal{N}(\mathcal{G}, \mathcal{I}_p).$$

If we assume that $h' \in \mathcal{H}$ satisfying $\sup_{h \in \mathcal{H}} \left| \mathcal{L}_{Z_{\mathcal{I}_p}}(h) - \sum_{i \in \mathcal{I}_p} \alpha_i \mathbb{E}[\ell(h, z_i)] \right| = \mathcal{L}_{Z_{\mathcal{I}_p}}(h') - \sum_{i \in \mathcal{I}_p} \alpha_i \mathbb{E}[\ell(h', z_i)]$, we have $\mathcal{L}_{Z_{\mathcal{I}_p}}(\hat{h}) - \mathcal{L}_{Z_{\mathcal{I}_p}}(\hat{h}^*) \leq b \sum_{i \in \mathcal{I}_p} \sum_{z_i \in \mathcal{Z}} P_{Z_i}^2 + \mathcal{N}(\mathcal{G}, \mathcal{I}_p)$.

To sum up, we eventually have

$$\mathcal{L}_{Z_{\mathcal{I}_p}}(\hat{h}) - \mathcal{L}_{Z_{\mathcal{I}_p}}(\hat{h}^*) \leq \mathcal{E}_p + cb\sqrt{\frac{VC(\mathcal{G})}{\sum_{i=1}^N n_i}} + b\sqrt{\frac{ln(1/\delta)}{2\sum_{i=1}^N n_i}},$$

where the term $\mathcal{E}_p$ represents $2b \sum_{i \in \mathcal{I}_p} \sum_{z_i \in \mathcal{Z}} P_{Z_i}^2$ derived in the above. $\qquad \square$

On the basis of the proven three lemmas introduced above, we can find that Theorem 1 is proven immediately. $\qquad \square$

### A.2 PROOF OF THEOREM 2

*Proof.* Following the proof of Theorem 1, we can also decompose and amplify the gap $|\mathcal{L}_{Z^{\mathcal{I}}}(\hat{h}_t^*) - \mathcal{L}_{Z_{\mathcal{I}_t}}(\hat{h}_t^*)|$ as,

$$|\mathcal{L}_{Z^{\mathcal{I}}}(\hat{h}_t^*) - \mathcal{L}_{Z^{\mathcal{I}_t}}(\hat{h}_t^*)| \leq \underbrace{\sup_{h \in \mathcal{H}} \left| \mathcal{L}_{Z^{\mathcal{I}}}(\hat{h}_t^*) - \mathcal{L}_{Z^{\mathcal{I}}}(h) \right|}_{\text{overfitting error}} + \underbrace{\left| \mathcal{L}_{Z^{\mathcal{I}}}(\hat{h}_t) - \mathcal{L}_{Z_{\mathcal{I}_t}}(\hat{h}_t) \right|}_{\text{participation gap}} + \underbrace{\mathcal{L}_{Z_{\mathcal{I}_t}}(\hat{h}_t) - \mathcal{L}_{Z_{\mathcal{I}_t}}(\hat{h}_t^*)}_{\text{semi-excess risk}}.$$

We can use the theoretical results derived in Theorem 1 to bound the above overfitting error and the semi-excess risk directly. Therefore, we mainly focus on the participation gap in the client selection scenario in the following. The participation gap can be decomposed and amplified as,

$$
\begin{aligned}
\left| \mathcal{L}_{Z^{\mathcal{I}}}(\hat{h}_t) - \mathcal{L}_{Z_{\mathcal{I}_t}}(\hat{h}_t) \right| &= \left| \sum_{\mathcal{Z}^{\mathcal{I}}} P_{Z^{\mathcal{I}}} \frac{1}{N} \sum_{i \in \mathcal{I}} \ell(\hat{h}_t, z_i) \log\left(\frac{1}{P_{Z^{\mathcal{I}}}}\right) - \frac{1}{K} \sum_{i \in \mathcal{I}_t} \sum_{z_i \in \mathcal{Z}} P_{Z_i} \ell(\hat{h}_t, z_i) \log\left(\frac{1}{P_{Z_i}}\right) \right| \\
&\leq \left| \sum_{\mathcal{Z}^{\mathcal{I}}} P_{Z^{\mathcal{I}}} \frac{1}{N} \sum_{i \in \mathcal{I}} \ell(\hat{h}_t, z_i) \log\left(\frac{1}{P_{Z^{\mathcal{I}}}}\right) - \sum_{\mathcal{Z}^{\mathcal{I}_p}} P_{Z^{\mathcal{I}_P}} \frac{1}{M} \sum_{i \in \mathcal{I}_p} \ell(\hat{h}_t, z_i) \log\left(\frac{1}{P_{Z^{\mathcal{I}_p}}}\right) \right| \\
&\quad + \left| \sum_{\mathcal{Z}^{\mathcal{I}_p}} P_{Z^{\mathcal{I}_p}} \frac{1}{M} \sum_{i \in \mathcal{I}_p} \ell(\hat{h}_t, z_i) \log\left(\frac{1}{P_{Z^{\mathcal{I}_p}}}\right) - \frac{1}{K} \sum_{i \in \mathcal{I}_t} \sum_{z_i \in \mathcal{Z}} P_{Z_i} \ell(\hat{h}_t, z_i) \log\left(\frac{1}{P_{Z_i}}\right) \right|.
\end{aligned}
$$

We move to the first term in the right hand side of the inequality. Similar to the proof of Theorem 1, we can easily have,

$$\Big| \sum_{\mathcal{Z}^{\mathcal{I}}} P_{Z^{\mathcal{I}}} \frac{1}{N} \sum_{i \in \mathcal{I}} \ell(\hat{h}_t, z_i) \log \Big( \frac{1}{P_{Z^{\mathcal{I}}}} \Big) - \sum_{\mathcal{Z}^{\mathcal{I}_p}} P_{Z^{\mathcal{I}_P}} \frac{1}{M} \sum_{i \in \mathcal{I}_p} \ell(\hat{h}_t, z_i) \log \Big( \frac{1}{P_{Z^{\mathcal{I}_p}}} \Big) \Big|$$

$$\leq \Big| \sum_{\mathcal{Z}^{\mathcal{I}}} P_{Z^{\mathcal{I}}} \frac{1}{N} \sum_{i \in \mathcal{I} \setminus \mathcal{I}_p} \ell(\hat{h}_t, z_i) \log \Big( \frac{1}{P_{Z^{\mathcal{I}}}} \Big) \Big| + \Big| \sum_{\mathcal{Z}^{\mathcal{I}}} P_{Z^{\mathcal{I}}} \frac{1}{N} \sum_{i \in \mathcal{I}_p} \ell(\hat{h}_t, z_i) \log \Big( \frac{1}{P_{Z^{\mathcal{I}}}} \Big)$$
$$- \sum_{\mathcal{Z}^{\mathcal{I}_p}} P_{Z^{\mathcal{I}_P}} \frac{1}{M} \sum_{i \in \mathcal{I}_p} \ell(\hat{h}_t, z_i) \log \Big( \frac{1}{P_{Z^{\mathcal{I}_p}}} \Big) \Big|$$

$$\leq \frac{b(N-M)}{N} H(Z^{\mathcal{I}}) + \Big| \sum_{\mathcal{Z}^{\mathcal{I}}} P_{Z^{\mathcal{I}}} \frac{1}{N} \sum_{i \in \mathcal{I}_p} \ell(\hat{h}_t, z_i) \log \Big( \frac{1}{P_{Z^{\mathcal{I}}}} \Big) - \sum_{\mathcal{Z}^{\mathcal{I}_p}} \sum_{\mathcal{Z}^{\mathcal{I} \setminus \mathcal{I}_p}} P_{Z^{\mathcal{I}_P}} P_{Z^{\mathcal{I} \setminus \mathcal{I}_P}} \frac{1}{M} \sum_{i \in \mathcal{I}_p} \ell(\hat{h}_t, z_i) \log \Big( \frac{1}{P_{Z^{\mathcal{I}_p}}} \Big) \Big|$$

$$\leq \frac{b(N-M)}{N} H(Z^{\mathcal{I}}) + \sum_{\mathcal{Z}^{\mathcal{I}}} P_{Z^{\mathcal{I}}} \Big| \frac{1}{N} \sum_{i \in \mathcal{I}_p} \ell(\hat{h}_t, z_i) \log \Big( \frac{1}{P_{Z^{\mathcal{I}}}} \Big) - \frac{1}{M} \sum_{i \in \mathcal{I}_p} \ell(\hat{h}_t, z_i) \log \Big( \frac{1}{P_{Z^{\mathcal{I}_p}}} \Big) \Big|$$

$$= \frac{b(N-M)}{N} H(Z^{\mathcal{I}}) + \sum_{\mathcal{Z}^{\mathcal{I}}} P_{Z^{\mathcal{I}}} \Big| [\frac{1}{N} - \frac{1}{M}] \sum_{i \in \mathcal{I}_p} \ell(\hat{h}_t, z_i) \log \Big( \frac{1}{P_{Z^{\mathcal{I}}}} \Big) \Big|$$

$$+ \sum_{\mathcal{Z}^{\mathcal{I}}} P_{Z^{\mathcal{I}}} \Big| \frac{1}{M} \sum_{i \in \mathcal{I}_p} \ell(\hat{h}_t, z_i) [\log \Big( \frac{1}{P_{Z^{\mathcal{I}}}} \Big) - \log \Big( \frac{1}{P_{Z^{\mathcal{I}_p}}} \Big)] \Big|$$

$$= \frac{b(N-M)}{N} H(Z^{\mathcal{I}}) + b \sum_{\mathcal{Z}^{\mathcal{I}}} P_{Z^{\mathcal{I}}} \sum_{i \in \mathcal{I}_p} \frac{N-M}{NM} \log \Big( \frac{1}{P_{Z^{\mathcal{I}}}} \Big) + b \sum_{\mathcal{Z}^{\mathcal{I}}} P_{Z^{\mathcal{I}}} \frac{1}{M} \sum_{i \in \mathcal{I}_p} \Big| \log \Big( \frac{1}{P_{Z^{\mathcal{I}}}} \Big) - \log \Big( \frac{1}{P_{Z^{\mathcal{I}_p}}} \Big) \Big|$$

$$= \frac{b(3N-2M)}{N} H(Z^{\mathcal{I}}) - b H(Z^{\mathcal{I}_P}).$$

According to the above analysis and Lemma 2, we can directly bound the second term and have,

$$\Big| \sum_{\mathcal{Z}^{\mathcal{I}_p}} P_{Z^{\mathcal{I}_P}} \frac{1}{M} \sum_{i \in \mathcal{I}_p} \ell(\hat{h}_t, z_i) \log \Big( \frac{1}{P_{Z^{\mathcal{I}_p}}} \Big) - \frac{1}{K} \sum_{i \in \mathcal{I}_t} \sum_{z_i \in \mathcal{Z}} P_{Z_i} \ell(\hat{h}_t, z_i) \log \Big( \frac{1}{P_{Z_i}} \Big) \Big|$$
$$\leq \frac{2b(M-K)}{M} H(Z^{\mathcal{I}_P}) + b H(Z^{\mathcal{I}_P}) - \frac{b}{K} \sum_{i \in \mathcal{I}_t} H(Z_i).$$

To sum up, we can derive the final participation gap in this scenario as follows,

$$\Big| \mathcal{L}_{Z^{\mathcal{I}}}(\hat{h}_t) - \mathcal{L}_{Z_{\mathcal{I}_t}}(\hat{h}_t) \Big| \leq \frac{b(3N-2M)}{N} H(Z^{\mathcal{I}}) + \frac{2b(M-K)}{M} H(Z^{\mathcal{I}_P}) - \frac{b}{K} \sum_{i \in \mathcal{I}_t} H(Z_i)$$

$$= \frac{b(3N-2M)}{N} H(Z^{\mathcal{I}}) + \frac{b(2MK-2K^2-M)}{MK} H(Z^{\mathcal{I}_P}) + \frac{b}{K} \Big[ H(Z^{\mathcal{I}_P}) - \sum_{i \in \mathcal{I}_t} H(Z_i) \Big]$$

$$= b(3 - \frac{2M}{N}) H(Z^{\mathcal{I}}) + b(2 - \frac{2K}{M} - \frac{1}{K}) H(Z^{\mathcal{I}_P}) + \frac{b}{K} \Big[ H(Z^{\mathcal{I}_P}) - \sum_{i \in \mathcal{I}_t} H(Z_i) \Big].$$

The term $\frac{b}{K} \Big[ H(Z^{\mathcal{I}_P}) - \sum_{i \in \mathcal{I}_t} H(Z_i) \Big]$ can be further derived as follows,

$$\frac{b}{K} \Big[ H(Z^{\mathcal{I}_P}) - \sum_{i \in \mathcal{I}_t} H(Z_i) \Big] \leq \frac{b}{K} \Big[ H(Z^{\mathcal{I}_P}) - H(Z^{\mathcal{I}_t}) \Big]$$

$$\leq \frac{b}{K} \Big[ H(Z^{\mathcal{I}_P}) - H(Z^{\mathcal{I}_t} | Z^{\mathcal{I}_P \setminus \mathcal{I}_t}) \Big]$$

$$= \frac{b}{K} \Big[ H(Z^{\mathcal{I}_P}) - H(Z^{\mathcal{I}_P} | Z^{\mathcal{I}_P \setminus \mathcal{I}_t}) \Big]$$

$$= \frac{b}{K} I(Z^{\mathcal{I}_P}; Z^{\mathcal{I}_P \setminus \mathcal{I}_t}).$$

The first inequality holds since the chain rule property of entropy and the second inequality holds since conditioning reduces entropy. The mutual information $I(Z^{\mathcal{I}_p}; Z^{\mathcal{I}_p \backslash \mathcal{I}_t})$ can be rewritten by the form of KL-divergence, i.e., $I(Z^{\mathcal{I}_p}; Z^{\mathcal{I}_p \backslash \mathcal{I}_t}) = KL(P_{Z^{\mathcal{I}_p}} \| P_{Z^{\mathcal{I}_p}} P_{Z^{\mathcal{I}_p \backslash \mathcal{I}_t}})$, we thus have,

$$
\begin{aligned}
\frac{b}{K}\Big[H(Z^{\mathcal{I}_p}) - \sum_{i \in \mathcal{I}_t} H(Z_i)\Big] &\le \frac{b}{K} KL(P_{Z^{\mathcal{I}_p}} \| P_{Z^{\mathcal{I}_p}} P_{Z^{\mathcal{I}_p \backslash \mathcal{I}_t}}) \\
&= \frac{b}{K} \sum_{\mathcal{Z}^{\mathcal{I}_p}} P_{Z^{\mathcal{I}_p}} \log\Big(\frac{P_{Z^{\mathcal{I}_p}}}{P_{Z^{\mathcal{I}_p}} P_{Z^{\mathcal{I}_p \backslash \mathcal{I}_t}}}\Big) \\
&= \frac{b}{K} \sum_{\mathcal{Z}^{\mathcal{I}_p \backslash \mathcal{I}_t}} P_{Z^{\mathcal{I}_p \backslash \mathcal{I}_t}} \log\Big(\frac{1}{P_{Z^{\mathcal{I}_p \backslash \mathcal{I}_t}}}\Big) \\
&= \frac{b}{K} H(Z^{\mathcal{I}_p \backslash \mathcal{I}_t}) \\
&\le \frac{b}{K} \sum_{i \in \mathcal{I}_p \backslash \mathcal{I}_t} H(Z_i) \\
&\le \frac{b}{K} \sum_{i \in \mathcal{I}_p \backslash \mathcal{I}_t} H(P_{Z_i}, P_{Z_j}), \forall j \ne i, i, j \in \mathcal{I}.
\end{aligned}
$$

Rooted on the results obtained in Theorem 1, we can similarly have the following upper bound of generalization gap in the client selection scenario,

$$
\begin{aligned}
\mathcal{L}_{Z^{\mathcal{I}}}(\hat{h}_t^*) - \mathcal{L}_{Z_{\mathcal{I}_t}}(\hat{h}_t^*) &\le L\|\hat{h}_t^* - h_t^*\| H(Z^{\mathcal{I}}) + b(3 - \frac{2M}{N}) H(Z^{\mathcal{I}}) + b(2 - \frac{2K}{M} - \frac{1}{K}) H(Z^{\mathcal{I}_p}) \\
&\quad + \frac{b}{K} \sum_{i \in \mathcal{I}_p \backslash \mathcal{I}_t} H(P_{Z_i}, P_{Z_j}) + \mathcal{E}_t + cb\sqrt{\frac{VC(\mathcal{G})}{\sum_{i=1}^{|\mathcal{I}_t|} n_i}} + b\sqrt{\frac{\log(1/\delta)}{2\sum_{i=1}^{|\mathcal{I}_t|} n_i}}.
\end{aligned}
$$

where $c$ is a constant. $h^* := \sup_{h \in \mathcal{H}} L\|\hat{h}^* - h\| H(Z^{\mathcal{I}})$ and $L$ is the Lipschitz constant. $\mathcal{E}_t = 2b \sum_{i \in \mathcal{I}_t} \sum_{z_i \in \mathcal{Z}} P_{Z_i}^2$. $H(P_{Z_i}, P_{Z_j})$ is the cross entropy between distributions $P_{Z_i}$ and $P_{Z_j}$. $\qquad \square$

## B  GENERALIZATION ANALYSIS OF IN-DISTRIBUTION SCENARIO

In this section, we will add the detail of generalization analysis in terms of the in-distribution scenario that all the clients participate in FL, i.e., $\mathcal{I}_p = \mathcal{I}$ and thus $N = M$. We will show that the average participation gap is related to the entropy rate of stochastic process $\{Z_i\}_{i \in \mathcal{I}}$.

Recall that the participation gap defined in this paper is as follows,

$$
\Big|\mathbb{E}_{Z^{\mathcal{I}}}\Big[\frac{1}{N}\sum_{i \in \mathcal{I}} \ell(\hat{h}, Z_i) \log\Big(\frac{1}{P_{Z^{\mathcal{I}}}}\Big)\Big] - \sum_{i \in \mathcal{I}_p} \alpha_i \mathbb{E}_{Z_i}\Big[\ell(\hat{h}, Z_i) \log\Big(\frac{1}{P_{Z_i}}\Big)\Big]\Big| \tag{13}
$$

Now we assume all the possible clients participate in FL and the server selects all the participating clients in each round of FL, therefore we have $\mathcal{I}_p = \mathcal{I}$ and $N = M$. Eventually, the participation gap in this scenario becomes a generalization gap from the discrepancy between distributed learning and centralized training:

$$
gen(Z^{\mathcal{I}}, \hat{h}) := \Big|\mathbb{E}\Big[\sum_{i \in \mathcal{I}} \alpha_i \ell(\hat{h}, Z_i) \log\frac{1}{P_{Z^{\mathcal{I}}}}\Big] - \sum_{i \in \mathcal{I}} \alpha_i \mathbb{E}\Big[\ell(\hat{h}, Z_i) \log\frac{1}{P_{Z_i}}\Big]\Big|
$$

**Theorem 3.** *Assume $\ell(w, Z)$ is bounded by $b$.*

$$
gen(Z^{\mathcal{I}}, \hat{h}) = \Big|\mathbb{E}\Big[\sum_{i \in \mathcal{I}} \alpha_i \ell(\hat{h}, Z_i) \log\frac{1}{P_{Z^{\mathcal{I}}}}\Big] - \sum_{i \in \mathcal{I}} \alpha_i \mathbb{E}\Big[\ell(\hat{h}, Z_i) \log\frac{1}{P_{Z_i}}\Big]\Big| \le bH(Z^{\mathcal{I}}) - b \sum_{i \in \mathcal{I}} \alpha_i H(Z_i) \tag{14}
$$

*Proof.*

$$gen(Z^{\mathcal{I}}, \hat{h}) = \left| \mathbb{E}\Big[ \sum_{i \in \mathcal{I}} \alpha_i \ell(\hat{h}, Z_i) \log \frac{1}{P_{Z^{\mathcal{I}}}} \Big] - \sum_{i \in \mathcal{I}} \alpha_i \mathbb{E}\Big[ \ell(\hat{h}, Z_i) \log \frac{1}{P_{Z_i}} \Big] \right|$$

$$= \left| \sum_{i \in \mathcal{I}} \alpha_i \mathbb{E}\Big[ \ell(\hat{h}, Z_i) \log \frac{1}{P_{Z^{\mathcal{I}}}} \Big] - \sum_{i \in \mathcal{I}} \alpha_i \mathbb{E}\Big[ \ell(\hat{h}, Z_i) \log \frac{1}{P_{Z_i}} \Big] \right|$$

$$\leq \sum_{i \in \mathcal{I}} \alpha_i \left| \mathbb{E}\Big[ \ell(\hat{h}, Z_i) \log \frac{1}{P_{Z^{\mathcal{I}}}} \Big] - \mathbb{E}\Big[ \ell(\hat{h}, Z_i) \log \frac{1}{P_{Z_i}} \Big] \right|$$

$$= \sum_{i \in \mathcal{I}} \alpha_i \left| \sum_{\mathcal{Z}^{\mathcal{I}}} P_{Z^{\mathcal{I}}} \ell(\hat{h}, z) \log \frac{1}{P_{Z^{\mathcal{I}}}} - \sum_{\mathcal{Z}} P_{Z_i} \ell(\hat{h}, z) \log \frac{1}{P_{Z_i}} \right|$$

$$= \sum_{i \in \mathcal{I}} \alpha_i \left| \sum_{\mathcal{Z}^{\mathcal{I}}} P_{Z^{\mathcal{I}}} \ell(\hat{h}, z) \log \frac{1}{P_{Z^{\mathcal{I}}}} - \sum_{\mathcal{Z}} \sum_{\mathcal{Z}^{\mathcal{I} \setminus i}} P_{Z_i} P_{Z^{\mathcal{I} \setminus i}} \ell(\hat{h}, z) \log \frac{1}{P_{Z_i}} \right|$$

$$= \sum_{i \in \mathcal{I}} \alpha_i \left| \sum_{\mathcal{Z}^{\mathcal{I}}} P_{Z^{\mathcal{I}}} \Big[ \ell(\hat{h}, z) \log \frac{1}{P_{Z^{\mathcal{I}}}} - \ell(\hat{h}, z) \log \frac{1}{P_{Z_i}} \Big] \right|$$

$$\leq \sum_{i \in \mathcal{I}} \alpha_i \sum_{\mathcal{Z}^{\mathcal{I}}} P_{Z^{\mathcal{I}}} \left| \ell(\hat{h}, z) \log \frac{1}{P_{Z^{\mathcal{I}}}} - \ell(\hat{h}, z) \log \frac{1}{P_{Z_i}} \right|$$

$$\leq b \sum_{i \in \mathcal{I}} \alpha_i \sum_{\mathcal{Z}^{\mathcal{I}}} P_{Z^{\mathcal{I}}} \left| \log \frac{1}{P_{Z^{\mathcal{I}}}} - \log \frac{1}{P_{Z_i}} \right|$$

$$= b \sum_{i \in \mathcal{I}} \alpha_i \sum_{\mathcal{Z}^{\mathcal{I}}} P_{Z^{\mathcal{I}}} \Big[ \log \frac{1}{P_{Z^{\mathcal{I}}}} - \log \frac{1}{P_{Z_i}} \Big]$$

$$= b H(Z^{\mathcal{I}}) - b \sum_{i \in \mathcal{I}} \alpha_i H(Z_i)$$

$\square$

**Corollary 1.** *We denote the cardinality of the considered sample space $\mathcal{Z}$ as $|\mathcal{Z}|$. Let the weighting factor $\alpha_i$ be $\frac{1}{N}$ for each client, we have,*

$$\left| \mathbb{E}\Big[ \sum_{i \in \mathcal{I}} \frac{\ell(\hat{h}, Z_i)}{N} \log \frac{1}{P_{Z^{\mathcal{I}}}} \Big] - \frac{1}{N} \sum_{i \in \mathcal{I}} \mathbb{E}\Big[ \ell(\hat{h}, Z_i) \log \frac{1}{P_{Z_i}} \Big] \right| \leq b(N-1) \log |\mathcal{Z}| \tag{15}$$

*Remark* 3. Corollary 1 states that in the in-distribution scenario defined above, the cardinality of $\mathcal{Z}$ and the number of clients impact the generalization performance of FL.

*Proof.* Similarly, referring to the proof of Theorem 3, we can have,

$$\left| \mathbb{E}\Big[ \sum_{i \in \mathcal{I}} \frac{\ell(\hat{h}, Z_i)}{N} \log \frac{1}{P_{Z^{\mathcal{I}}}} \Big] - \frac{1}{N} \sum_{i \in \mathcal{I}} \mathbb{E}\Big[ \ell(\hat{h}, Z_i) \log \frac{1}{P_{Z_i}} \Big] \right| \leq b\Big[ H(Z^{\mathcal{I}}) - \frac{1}{N} \sum_{i \in \mathcal{I}} H(Z_i) \Big]$$

$$\leq b\Big[ \sum_{i \in \mathcal{I}} H(Z_i) - \frac{1}{N} \sum_{i \in \mathcal{I}} H(Z_i) \Big]$$

$$\leq \frac{b(N-1)}{N} \sum_{i \in \mathcal{I}} H(Z_i)$$

$$\leq \frac{b(N-1)}{N} \sum_{i \in \mathcal{I}} \log |\mathcal{Z}|$$

$$= b(N-1) \log |\mathcal{Z}| \tag{16}$$

$\square$

Furthermore, we assign the weighting factor $\alpha_i = \alpha_j = \frac{1}{N}, \forall i, j \in \mathcal{I}$ for each client and we define the average participation gap as,

**Definition 4** (Averaged generalization gap between distributed and centralized training)**.**

$$\frac{1}{N}\Big|\mathbb{E}_{Z^{\mathcal{I}}}\Big[\frac{1}{N}\sum_{i\in\mathcal{I}}\ell(\hat{h},Z_i)\log\Big(\frac{1}{P_{Z^{\mathcal{I}}}}\Big)\Big]-\frac{1}{N}\sum_{i\in\mathcal{I}}\mathbb{E}_{Z_i}\Big[\ell(\hat{h},Z_i)\log\Big(\frac{1}{P_{Z_i}}\Big)\Big]\Big|$$

*Remark* 4. This definition represents the average generalization gap between distributed and centralized training for each participating clients.

**Theorem 4.** *Assume the entropy rate $H(\mathcal{Z}) = \lim_{N\to\infty}\frac{1}{N}H(Z^{\mathcal{I}})$ of stochastic process $\{Z_i\}_{i\in\mathcal{I}}$ exists. Let the weighting factor $\alpha_i$ be $\frac{1}{N}$ for each client, we have,*

$$\lim_{N\to\infty}\frac{1}{N}\Big|\mathbb{E}\Big[\sum_{i\in\mathcal{I}}\frac{\ell(\hat{h},Z_i)}{N}\log\frac{1}{P_{Z^{\mathcal{I}}}}\Big]-\frac{1}{N}\sum_{i\in\mathcal{I}}\mathbb{E}\Big[\ell(\hat{h},Z_i)\log\frac{1}{P_{Z_i}}\Big]\Big|\leq bH(\mathcal{Z}) \quad (17)$$

*Remark* 5. Theorem 4 indicates that if the entropy rate $H(\mathcal{Z})$ of $\{Z_i\}_{i\in\mathcal{I}}$ exists, it can serve as an upper bound for the average generalization gap defined above, which essentially represent that the average information rate or average uncertainty associated with the considered stochastic process $\{Z_i\}_{i\in\mathcal{I}}$ influences the generalization performance of FL.

*Proof.* Following the proof of Theorem 3, we can have,

$$\frac{1}{N}\Big|\mathbb{E}\Big[\sum_{i\in\mathcal{I}}\frac{\ell(\hat{h},Z_i)}{N}\log\frac{1}{P_{Z^{\mathcal{I}}}}\Big]-\frac{1}{N}\sum_{i\in\mathcal{I}}\mathbb{E}\Big[\ell(\hat{h},Z_i)\log\frac{1}{P_{Z_i}}\Big]\Big|\leq\frac{b}{N}[H(Z^{\mathcal{I}})-\frac{1}{N}\sum_{i\in\mathcal{I}}H(Z_i)]$$

$$\leq\frac{b}{N}[H(Z^{\mathcal{I}})-\frac{1}{N}H(Z^{\mathcal{I}})]$$

$$=\frac{bH(Z^{\mathcal{I}})}{N}-\frac{1}{N}\frac{bH(Z^{\mathcal{I}})}{N} \quad (18)$$

Since the limit $\lim_{N\to\infty}\frac{H(Z^{\mathcal{I}})}{N}$ exists and it is denoted by $H(\mathcal{Z})$, then we can have,

$$\lim_{n\to\infty}\frac{1}{N}\Big|\mathbb{E}\Big[\sum_{i\in\mathcal{I}}\frac{\ell(\hat{h},Z_i)}{N}\log\frac{1}{P_{Z^{\mathcal{I}}}}\Big]-\frac{1}{N}\sum_{i\in\mathcal{I}}\mathbb{E}\Big[\ell(\hat{h},Z_i)\log\frac{1}{P_{Z_i}}\Big]\Big|\leq\lim_{N\to\infty}\Big\{\frac{bH(Z^{\mathcal{I}})}{N}-\frac{1}{N}\frac{bH(Z^{\mathcal{I}})}{N}\Big\}$$

$$=bH(\mathcal{Z}) \quad (19)$$

$\square$

# C  DETAILS OF METHODS

## C.1  EMPIRICAL ENTROPY-BASED WEIGHTING

Recall that this paper only focuses on the label distribution skew scenario for verifying the proposed empirical entropy-based weighting method. In other words, the aggregation weighting factor can be designed as follows:

$$\alpha_i=\frac{\exp(\hat{H}_i)}{\sum_{i\in\mathcal{I}_p}\exp(\hat{H}_i)}. \quad (20)$$

where $\hat{H}_i = -\sum_{y\in\mathcal{Y}}\frac{\sum_j\mathbb{I}_{y=y_j^i}}{n_i}\log\frac{\sum_j\mathbb{I}_{y=y_j^i}}{n_i}$, $y_j^i$ is the label of $j$-th sample of local dataset $S_i$ of client $i$ and $\mathbb{I}$ denotes the indicator function.

The detailed workflow of the proposed empirical entropy-based weighting method is introduced as follows. Before the start of FL, each participating client is required to calculate the empirical entropy in equation 20 based on its local dataset and further uploads the empirical entropy to the server. The server can thus assign the aggregating weighting factors for all the clients rooted on the received empirical entropy prior to the first round of FL.

**Privacy Computations and Communication Costs:** Privacy computation techniques can be be integrated into the proposed methods: clients upload the empirical entropy of local data sources via

homomorphic encryption or secure multi-party computation with low computation overhead since the empirical entropy is a scalar value. We assume that data sources of participating clients are stationary during the whole FL process and hence clients only need to upload the empirical entropy to the server before FL starts only for one time, which can reduce the communication cost.

## C.2 Gradient similarity-based Client selection

Recall that the workflow of the client selection stage discussed in this paper. In general, the server iterates through the following steps in each round of FL:

- **(Update the gradient table):** After updating the global model based on local gradients uploaded by clients $i, \forall i \in \mathcal{I}_t$ following the update rule of $\mathbf{w}_{t+1} \leftarrow \mathbf{w}_t - \sum_{i \in \mathcal{I}_t} \alpha_i \mathbf{g}_t^i$, the server updates and maintains a table that stores the latest local gradients uploaded by the clients selected in each round. More specifically, the server performs the following actions: $\mathbf{g}_s^i \leftarrow \mathbf{g}_t^i, \forall i \in \mathcal{I}_t$ and $\mathbf{g}_s^i \leftarrow \mathbf{g}_s^i, \forall i \in \mathcal{I}_p \setminus \mathcal{I}_t$ to maintain the table $\{\mathbf{g}_s^i\}_{i \in \mathcal{I}_P}$.
- **(Execute the selection algorithm):** The server applies the client selection algorithms described below, utilizing the local gradients stored in the gradient table $\{\mathbf{g}_s^i\}_{i \in \mathcal{I}_p}$ in order to determine the clients $\mathcal{I}_{t+1}$ to participate in the next round of FL.

We then describe the detail procedures of two client selection methods proposed in this paper.

### C.2.1 Minimax gradient similarity-based client selection

We now provide the formal procedure of the proposed Minimax gradient similarity-based client selection in the following:

- **(Constructing the similarity set):** First, we develop a "similarity set" $\mathbb{S}_i$ for each stored gradients $\mathbf{g}_s^i, \forall i \in \mathcal{I}_p$. This "similarity set" $\mathbb{S}_i$ of $i$-th stored gradients $\mathbf{g}_s^i$ contains the cosine similarities $S_{i,j} = \frac{<\mathbf{g}_s^i, \mathbf{g}_s^j>}{\|\mathbf{g}_s^i\| \|\mathbf{g}_s^j\|}$ between $\mathbf{g}_s^i$ and all the other stored gradients $\mathbf{g}_s^j, \forall j \neq i, j \in \mathcal{I}_p$, i.e., $\mathbb{S}_i = \{S_{i,j}\}_{j \in \mathcal{I}_p, j \neq i}$.
- **(Calculating the maximum similarity):** Next, the server builds another set $\mathbb{S}_{max}$ to store the maximum similarity $\max_{S_{i,j}} \mathbb{S}_i$ in each $\mathbb{S}_i, \forall i \in \mathcal{I}_p$ while this maximum similarity measures the degree of the similarity between one data source $i$ and other data sources.
- **(Selecting clients with the smallest maximum similarity ):** In the final step, the server determines the clients whose maximum similarity $\max_{S_{i,j}} \mathbb{S}_i$ is the smallest in the maximum similarity set $\mathbb{S}_{max}$.

The fundamental idea behind the aforementioned operations is to identify data sources that are "distant" from other data sources, thereby approximately achieving the objective stated in equation 11.

### C.2.2 Convex hull construction-based client selection

Prior to delving into the specifics of the client selection method based on convex hull construction, we will begin by providing the definition of the convex hull. The convex hull of a set $C$, denoted as **conv**$C$, refers to the set of all convex combinations of points in $C$ (Boyd and Vandenberghe, 2004):

$$\mathbf{conv}C := \{\lambda_1 x_1 + \lambda_1 x_2 + ... + \lambda_n x_n : x_i \in C, \lambda_i \geq 0, i = 1, 2, ..., n, \sum_{i=1}^{n} \lambda_i = 1\}. \tag{21}$$

Notice that the convex hull of point set $C$ represents the smallest convex set that encompasses all the points in $C$. The definition of the convex hull is visually depicted in Fig. 2.

The core idea of the proposed convex hull construction-based client selection is to identify the vertices of the convex hull formed by the stored gradients in the server's table. Subsequently, the clients whose gradients are located at these vertices of the constructed convex hull are selected.

The formal work process of the proposed convex hull construction-based client selection is outlined as follows. a) The server initiates the execution of the quickhull algorithm proposed in Barber et al.

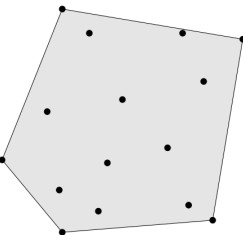

Figure 2: The convex hull of a point set in $\mathbb{R}^2$. The convex hull of a point set of 15 points is the pentagon (shown shaded).

(1996) to construct the convex hull of the local gradients stored in the gradient table $\{\mathbf{g}_s^i\}_{i\in\mathcal{I}_p}$. b) Once the vertices of the considered point set, which comprises all the gradients stored in $\{\mathbf{g}_s^i\}_{i\in\mathcal{I}_p}$ are identified, the server proceeds to select the corresponding clients whose gradients are located on these discovered vertices. These selected clients are then required to participate in the subsequent round of FL.

Intuitively, the distances between points located on the vertices of the convex hull and all other points tend to be larger. This observation served as inspiration for our client selection method. Local gradients generated by possible unparticipating clients can be considered as "random points" occurring within or around a given point set. By utilizing the vertices of the convex hull, we can more effectively "cover" these "random points", providing a geometric perspective to further explain our method.

From an alternative interpretation, any points in a point set $C$ can be linearly represented by the convex hull $\mathbf{conv}C$, which is the smallest convex set contains $C$. This implies that only the gradients located on the vertices of the constructed convex hull can linearly represent gradients of the entire set. In other words, the data sources generating gradients located on the vertices of the convex hull can effectively represent other data sources, thereby reducing the intrinsic information redundancy in FL.

## D    EXPERIMENTAL DETAILS

In this section, we fill in the details of the numerical experiments in Section 5. The convergence analysis of different weighting aggregation methods and client selection methods is carried out in the following.

### D.1    CONVERGENCE ANALYSIS OF WEIGHTING AGGREGATION METHODS

We first perform the convergence analysis on weighting aggregation methods for EMNIST-10 and CIFAR-10. The label distribution skew of FL is considered in this part. More specifically, we split the total training set into different clients via the Dirichlet distribution spitting. The splitting parameter $\alpha$ of the Dirichlet distribution is set as 0.1 and 0.05 for EMNIST-10 and CIFAR-10 respectively.

The convergence behavior of the proposed empirical entropy-based weighting method compared with other baselines for EMNIST-10 and CIFAR-10 is presented in Figure 3. For EMNIST-10, the proposed empirical entropy-based weighting method converges faster than the other two baselines and it also maintains the highest OOD test accuracy among these weighting methods after about 20-th communication round. For CIFAR-10, both the proposed empirical entropy-based weighting method and equality weighting method converge faster than the data size-based weighting method while the proposed weighting method converges more stably than other baselines. The above results show that giving higher aggregation weights to local gradients trained on data sources with greater information entropy will improve the generalization performance of the global model, which is matched with our theoretical basis in Theorem 1 comprehensively.

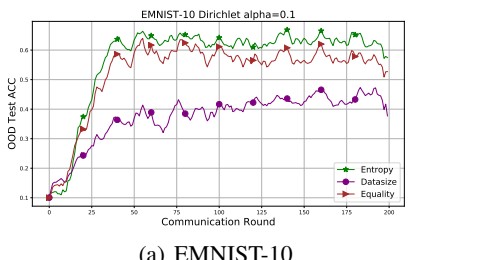 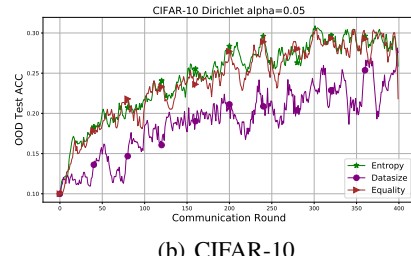

(a) EMNIST-10 (b) CIFAR-10

Figure 3: The convergence analysis on OOD test accuracy of empirical entropy weighting method compared with other weighting aggregation methods on EMNIST-10 and CIFAR-10.

## D.2 CONVERGENCE ANALYSIS OF CLIENT SELECTION METHODS

We then compare the convergence behavior of different client selection methods for the three datasets mentioned in Section 5. For EMNIST-10 and CIFAR-10, we still split the total training set via the Dirichlet distribution spitting. The splitting parameter $\alpha$ of the Dirichlet distribution is set as $0.5$ and $0.1$ for EMNIST-10 and CIFAR-10 respectively in this part. As mentioned in Section 5, for the Shakespeare dataset, each speaking role in each play is set as a local dataset.

We now focus on the convergence performance of the two proposed client selection methods in comparison with full sampling, power-of-choice selection, random selection and other proposed baselines for EMNIST-10 and Shakespeare in Figure 4.

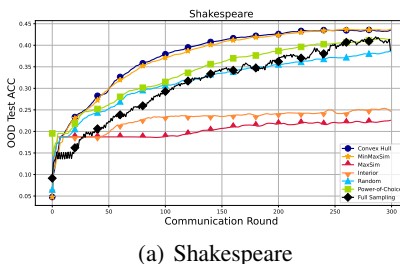 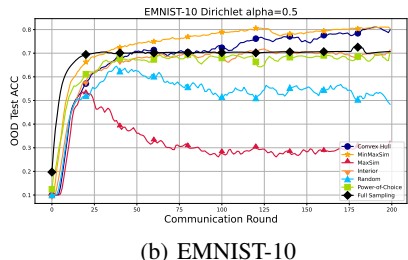

(a) Shakespeare (b) EMNIST-10

Figure 4: The convergence analysis on OOD test accuracy of two proposed client selection methods compared with random selection and other proposed baselines on Shakespeare and EMNIST-10.

For the Shakespeare dataset, we can find that the two proposed client selection methods (Minimax gradient similarity-based selection and convex hull construction-based selection) almost converge at the same rate and converge faster than full sampling, power-of choice selection, random selection and other baselines. For EMNIST-10, the Minimax gradient similarity-based selection can achieve the highest OOD test accuracy among all the client selection methods. In addition, it can be found that the full sampling scheme converges fastest and the most stably.

For Shakespeare and EMNIST-10 dataset, the reason why full sampling scheme achieves worse OOD test accuracy than the proposed selection methods may be that the randomness induced by the selection will even improve the out-of-distribution performance of the global model. From another perspective, the nature of the proposed methods is to "compress" the information from participating data sources, i.e., removing the redundant information from some data sources making contribution to OOD generalization less.

Besides, we also carry out experiments on the CIFAR-10 dataset for evaluating the convergence performance of the two proposed client selection methods and other baselines. The results in Figure 5 show that the two proposed client selection methods converge faster than their ablation baselines and random selection. And the two proposed methods converge more stably than power-of-choice selection. It indicates that selecting clients with more dissimilar local gradients and selecting clients

with local gradients in the convex hull not in the interior will both improve the generalization performance of the global model. According to Assumption 3 about the relationship between the gradient dissimilarity and the distribution discrepancy, we can get the following conclusion immediately: selecting clients with more diverse local distributions will improve the generalization performance of FL.

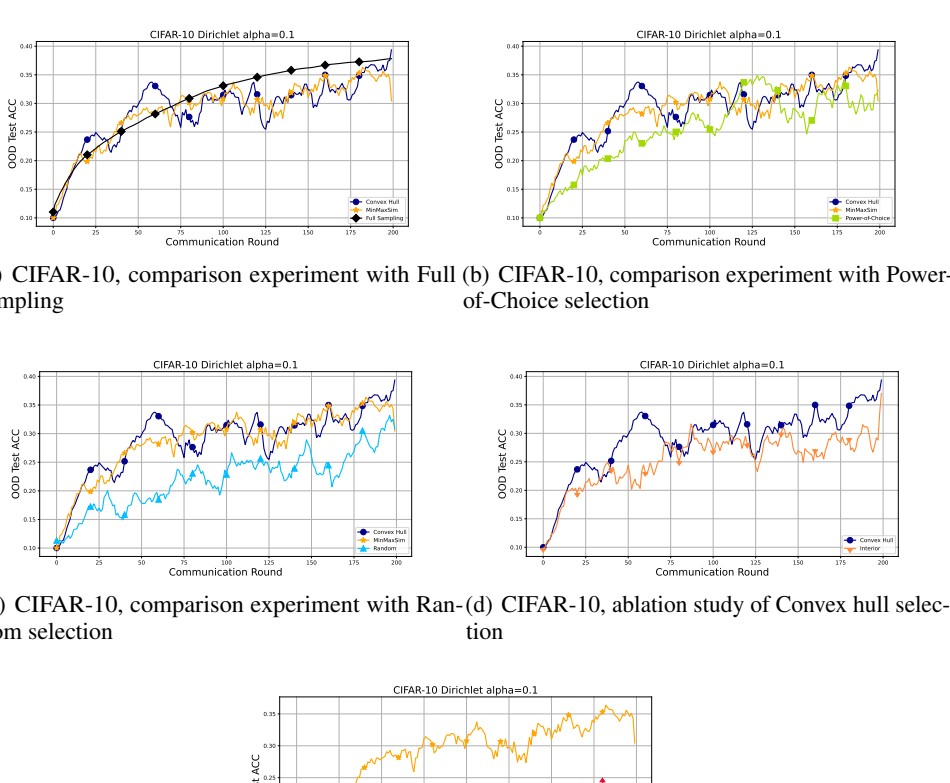

(a) CIFAR-10, comparison experiment with Full sampling

(b) CIFAR-10, comparison experiment with Power-of-Choice selection

(c) CIFAR-10, comparison experiment with Random selection

(d) CIFAR-10, ablation study of Convex hull selection

(e) CIFAR-10, ablation study of MinimaxSim selection

Figure 5: The convergence analysis on OOD test accuracy of two proposed client selection methods in comparison with Full sampling, Power-of-choice selection, random selection respectively and their ablation studies respectively.

On the other hand, the reason why the full sampling scheme performs more stably than all the client selection methods for CIFAR-10 is that training a model performing well on unseen distributions for CIFAR-10 is the most difficult among three tasks hence more participating clients will generalize the global model to unseen data source better.