# OpenReview forum: "Federated Generalization via Information-Theoretic Distribution Diversification"
_ICLR.cc/2024/Conference — Submitted to ICLR 2024_

### Official Review · Reviewer_tc4X · 2023-10-30

**Soundness:** 2 fair
**Presentation:** 3 good
**Contribution:** 1 poor
**Rating:** 3
**Confidence:** 3

**Summary:**

The paper investigates the effect of partial client participation in the training phase on generalization performance. The paper considers the non-i.i.d. (or heterogeneous) case, where all distributions of clients are different from each other. Moreover, instead of the classical generalization error, a new notion of "information-theoretic generalization gap" is introduced and studied. Following the established theoretical results, the authors proposed several variations of model aggregation in FL and showed the advantage of their methods through numerical simulations.

**Strengths:**

The generalization error in federated learning, and in particular the effect of client participation, is poorly understood theoretically. This work is one of the first to address this issue. Moreover, the proposed variations of FL aggregation seem to have some potential, as verified by experiments.

**Weaknesses:**

The main weaknesses of the paper are as follows (please see Questions for details).

- The considered setup does not well capture the FL setup

- The paper studies a newly defined "information-theoretic-generalization gap".  However, the concrete justification for studying that term is missing.

- The proof techniques are simple and the resulted bounds “seem” to be loose.

- Some related works are missing, including https://arxiv.org/abs/2303.01215, https://arxiv.org/abs/2304.12216, https://arxiv.org/abs/2306.05862, https://arxiv.org/abs/2306.05862.

For these reasons, I am unfortunately inclined to reject the paper. I would be willing to raise the score if the comments below can be addressed.

**Questions:**

It would be appreciated if the authors could clarify the following points.

1. One of the main weaknesses of the paper is that only the discrete alphabet is considered, which is not a realistic case. This limitation is mentioned only once in the Preliminary Section. However, it must also be emphasized in the abstract and the introduction. How the results would change in case of continuous alphabets and why this assumption is made?

2. Another major limitation is that, similar to some previous works, this work assumes that the FL algorithms used by the participating clients manage to find the "global minimizer" of the self-information weighted semi-empirical risk. This consequently means that the FL algorithm (considered for the participating clients), is "equivalent" to a centralized algorithm that has all training data of all participating clients in one place. In a sense, the paper assumes that there is no such thing as "distributed learning". While much of the work in the literature shows the different behavior of these architectures. In other words, it seems that the paper is not really about the federated learning setup. But rather the "mismatch" between test and training data.

3. My main concern is that, unless I have missed something, it seems that all results are in terms of the "information-theoretic-generalization gap". However, except for some "intuitions", it has not been shown concretely what is the relation between such a definition and the true "generalization error". In other words, what are the concrete implications of these results for the true generalization error? If this relation cannot be established, I am not sure about the usefulness of the results.

4. Considering Theorem 1, this result is obtained by almost purely algebraic manipulations (e.g., proofs of Lemmas 1 and 2) and excessive use of the triangle inequality. Besides the simplicity of the techniques used (except perhaps the proof of Lemma 3, which is inspired by and similar to previous work), the bounds using such techniques are susceptible to being very loose. Similar comments apply to Theorem 2. Therefore, considering the terms appearing in the bound as optimization proxies does not seem very justified.

5. The authors mentioned that “Our hypothesis is grounded in the notion that a model exhibiting proficiency with low-probability examples from training distributions might demonstrate adaptability to unfamiliar testing distributions.” In a sense, intuitively, the goal here is to consider the "worst case" scenarios. However, it seems to me that the information-theoretic approach is not appropriate for this goal. In essence, information theory relies heavily on the probabilistic behavior of the learned model and data distribution, and the associated concentration of measures. Could the authors intuitively justify their information-theoretic approach and consider "self-information-weighted expected risk"?

6. I think there are some mistakes in the definition of "semi-empirical risk" of Hu et al., 2023, just before section 3.2. In fact, they never considered "self-information-weighted semi-empirical risk" and in that paper $\hat{h}^* = \text{arg inf } \mathcal{L}_{\mathcal{D}}$. Do authors change their definitions to conform to the new measure "information-theoretic-generalization gap"? If so, what is the relationship to what was originally defined there?

7.  Describing the first term in the RHS of equation (4) as overfitting could be misleading. In fact, if, for example, $\hat{h}^*$ is the minimizer of the self-information weighted expected risk for the set $\mathcal{I}$, this term can still be large. What exactly is the intuition behind this term?

8.  Remark 1 simply rewrites the RHS of equation (5). But could you please give some intuition about the behavior described and why it makes sense?

---

> ### Author Response · Authors · 2023-11-15
> **Response to Reviewer tc4X**
>
> We thank the reviewer for the time to read our paper and the reviewer's hard work on our paper.
>
> **Questions**:
>
> * **About the discrete alphabet.** Thanks for your constructive comments! In fact, our paper adheres to the classic settings of Shannon information theory, specifically focusing on the discrete alphabet setup. Firstly, we emphasize that the information entropy of data sources serves as an upper bound for the proposed information-theoretic generalization gap. If we were to consider the continuous alphabets setting, the information entropy would be replaced by the concept of differential entropy. However, it is important to note that the differential entropy may assume negative values in certain cases, which can pose challenges when developing algorithms in such situations.
>
> * **About the distributed learning.** The Joint Self-Information Weighted Expected Risk, as defined in Definition 2, is equivalent to the scenario where all the data sources are stored in a centralized parameter server. In this setup, the server conducts centralized training using these data sources, as the expectation is computed based on the joint distribution of these data sources. Consequently, the Information-Theoretic Generalization Gap, defined in Definition 3, essentially quantifies the disparity between the information-theoretic expected risks calculated on the joint distribution of data sources and the marginal distribution of data sources. This gap signifies, to some extent, the difference between centralized learning and distributed learning (when $\mathcal{I}=\mathcal{I}_p$. i.e., all the clients participate in FL).
>
> * **About the connection between the true generalization error and the information theoretic generalization gap.** Thanks for your valuable comments! If we consider fixing the hypothesis $h \in \mathcal{H}$ and analyze the true generalization error and the proposed information-theoretic generalization gap using the Hoeffding inequality, we can observe that with high probability $(1-\delta)$, the true generalization error is bounded as $|\frac{1}{n}\sum_{i=1}^n\ell(h,z_i)-\mathbb{E}[\mathcal{L}(h)]|\leq\sqrt{\frac{b^2\log(2/\delta)}{2n}}$. Additionally, the information-theoretic generalization gap is bounded as $|\frac{1}{n}\sum_{i=1}^n\ell(h,z_i)\log\frac{1}{p(z_i)}-\mathbb{E}\ell(h,z)\log\frac{1}{p(z)}|\leq\sqrt{\frac{{(b\log(1/p_m))}^2\log(2/\delta)}{2n}}$, where $\ell$ is bounded by $b$, and $p_m$ represents the minimum probability of an outcome in $\mathcal{Z}$, with the condition $p_m\neq0$. The above bounds indicate that the information-theoretic generalization gap is dominated when there are outcomes with extremely high or low probabilities.
>
> * **About the techniques used in this paper.** The techniques employed in this paper are indeed based on general generalization theory and information theory. On one hand, for instance, in the proof of Theorem 2, we utilize techniques commonly employed in the field of information theory. Specifically, on page 19 of the supplementary material, we apply the concept of “conditioning reduces entropy” and leverage the properties of conditional entropy and mutual information to establish our proof. Additionally, in the analysis of the In-domain scenario, we utilize the entropy rate of a stochastic process to assess the complexity of the data sources and its influence on the generalization gap. Therefore, the techniques utilized in this paper are not limited to “purely algebraic manipulations.” Furthermore, the scope of this paper focuses on developing algorithms to enhance the out-of-distribution generalization of FL based on the derived generalization bound. This implies that the proposed generalization framework serves as a means to provide us with intuition and insights into understanding the OOD generalization of FL. Refining the presented generalization bound further is an avenue for our future research.

---

> > ### Author Response · Authors · 2023-11-15
> > **Response to Reviewer tc4X**
> >
> > * **About "the information-theoretic approach is not appropriate for this goal".** In fact, the scenario considered in this paper is commonly encountered in practice and does not represent the "worst case" scenario. For example, in the AIoT scenario, certain IoT devices may only collect data in specific areas, while the global model trained by these devices is expected to provide spatial-related services for all devices across the entire area. Consequently, certain outcomes within the sample space may have a relatively low probability within the training domain, but exhibit a higher probability within the test domain. During this period, the test domain is assumed to possess maximum entropy distributions, indicating a greater degree of uncertainty, as mentioned on Page 3 of the paper. It is important to note that considering maximum entropy distributions with significant uncertainty is a natural approach within the field of information theory.
> >
> > * **About  the mistakes in the definition of "semi-empirical risk".**  Thank you for your timely reminder. The definition of the "semi-empirical risk" in (Hu et al., 2023) is in fact given as $\frac{1}{m}\sum_{i}\mathbb{E}_{z \sim P_i} [\ell(h,z)]$, without taking into account the self-information of the outcome. However, in this paper, the term "semi-empirical risk" specifically refers to the self-information-weighted semi-empirical risk. We have adopted this metric based on the motivation provided by (Hu et al., 2023), and we have made the necessary modifications to reflect this in the latest version of the paper.
> >
> > * **About  the intuition behind the first term in the RHS of equation (4).** Thank you for your valuable comments! The first term $\sup_{h \in \mathcal{H}}L\Vert\hat{h}^*-h\Vert H(Z^{\mathcal{I}})$ in the upper bound in Theorem 1 indicates the distance between the optimal hypothesis $\hat{h}^*=\arg\inf_{h}\mathcal{L}_{\mathcal{I}_p}(h)$ on self-information-weighted semi-empirical risk and other $h$ in the hypothesis space. Therefore, this first term essentially reflects the effect of model complexity on the considered generalization gap and the difficulty in obtaining this optimal model.
> >
> > * **About  the Remark 1.** The key point in Equation 5 is the participation gap term $3bH(Z^{\mathcal{I}})-b\sum_{i}\alpha_iH(Z_i), i \in \mathcal{I}_p$. This term signifies that a smaller joint entropy of all possible data sources and a larger information entropy of participating data sources can lead to a reduction in the derived generalization bound. This observation is in line with our intuition. In our setting, we plan to train a global model by partial participating clients. This trained global model is intended to provide services for all possible clients. Consequently, the corresponding test domain becomes "complex" and generalizing the trained model to this test domain becomes challenging when the joint entropy $H(Z^{\mathcal{I}})$ is large. On the other hand, if the information entropy of the training domain is large, it implies that the trained model is powerful and capable of performing well on unknown test domains. This is because a larger information entropy indicates that the model has encountered a sufficient number of outcomes during the training stage.

---

> ### Comment · Reviewer_tc4X · 2023-11-17
>
> I would like to thank the authors for their time in writing the rebuttal. Unfortunately, I am not convinced that the paper at this stage is ready for publication. I think the paper has valuable ideas that can be further investigated for future submission.
>
> -  **Discrete Alphabet:** I’m not sure to agree about this. In information theory, indeed continuous alphabets (for both source and channel coding) have been vastly explored. Besides, concerning the source coding part, while the lossless compressibility for the discrete alphabet is measured by Shannon discrete entropy, the counterpart for the continuous sources is measuring the lossy compressibility via rate-distortion theory (and not differential entropy).
>
> -  **Distributed setup:** After clarification, unless I missed something, there seems to be another issue:  there does not exist any dependence on the training dataset in Definition 3, and calling this a generalization gap is not appropriate. Moreover, the dependence of the bounds (e.g. Theorem 1) on the dataset size is purely synthetic. Then, why one cannot let $n_i \to \infty$ and get rid of the last two terms?
>
> - **In connection with the generalization gap:** We cannot compare two terms by comparing their corresponding upper bounds.
>
> - **Regarding suitability of information-theoretic analysis:** Just to make it precise: exactly what I meant by “worst case” is considering the maximum entropy distributions, which e.g. for the finite alphabet case result in uniform distributions. Adhering to this principle, why in Definition 3, we do not consider such distributions for the set $Z^I$. In addition, why do we have still the term $\log(1/(P(Z^I)$? The justification is that this term for the “participating clients” is supposed to guarantee a good performance for all clients (for arbitrary distributions). But why we have $\log(1/(P(Z^I)$?

---

> ### Author Response · Authors · 2023-11-17
> **Response to Reviewer tc4X**
>
> Thanks for your detailed reply. Here is my further explanation.
>
> **Discrete Alphabet:** The key point of rate distortion theory is quantization, and the physical meaning of differential entropy is to measure the “difficulty” or the “degree of adjustment” involved in quantizing continuous sources. The vanilla definition of information entropy is unable to accurately measure the uncertainty of continuous sources. Therefore, we firmly believe that utilizing a discrete alphabet directly is sufficient to support our theoretical findings and uphold the spirit of Shannon’s information theory.
>
> **Distributed setup:** In our setup, the term “generalization” encompasses both “out-of-sample generalization” (which focuses on concentration) and “out-of-distribution generalization”. However, our primary focus in this paper is on the latter aspect. Equation 3 effectively quantifies the disparity between the information obtained from the entire data source  $Z^{\mathcal{I}}$  and the partial participating data sources  $Z^{\mathcal{I}_p}$. Therefore, in the context of Equation 3, the term “generalization” refers to the process of generalizing the model trained on  $Z^{\mathcal{I}_p}$ to the unknown $Z^{\mathcal{I}}$. The sample complexity term measures the impact of “out-of-sample generalization”, which tends to diminish as the size of the total samples stored in the participating training set approaches infinity.
>
> **In connection with the generalization gap:** By examining their respective upper bounds, we observe that in order to achieve the same approximate error $\epsilon$ , the necessary sample size $n$ for the self-information-weighted generalization gap is greater than that for the true generalization error,  when there are outcomes with extremely high or low probabilities.
>
> **Regarding suitability of information-theoretic analysis:** Thank you for providing further clarification. In fact, we formulated Definition 3 to ensure consistency in the expected risk form between the joint distribution-based joint self-information weighted expected risk defined in Equation 2 and the marginal distribution-based self-information weighted expected risk defined in Equation 1. This alignment of definitions aims to ensure that all of them are formulated based on a unified information-theoretic framework.

---

> > ### Comment · Reviewer_tc4X · 2023-11-17
> >
> > Thanks. I am not yet convinced about any of the above points, except the second aspect of using the word "generalization" which I think needs to be clearly stated in the paper since the term "participation gap" is used usually for this. I will think about your arguments.
> >
> > Just to clarify one point again: Definition 3 does not depend on the size of the training dataset ($n_i$), since $\hat{h}^*$ is the minimizer with respect to the weighted expectation terms. Am I right about this? If so, then this holds for "any" $n_i$, which means we can choose it as infinite. Then the last two terms in Theorem 1 are zero. Is this correct?

---

> > > ### Author Response · Authors · 2023-11-17
> > > **Response to Reviewer tc4X**
> > >
> > > Thank you for your timely reply.
> > >
> > >
> > >
> > > I understand your point. In fact, if $\sum_{i \in \mathcal{I}_p} n_i$ is infinite, the last two terms become zero based on the classical generalization theory, commonly referred to as “out-of-sample generalization,” as mentioned in the previous response. However,  if we aim to generalize the model trained on partially participating data sources to unknown test domains, there will still be a generalization gap. This is because the disparity between the source domains and the target domains will not diminish, leading to challenges in achieving optimal performance on unseen distributions.

---

> ### Comment · Reviewer_tc4X · 2023-11-23
>
> This does not answer my question. As I understand it, the object defined in Definition 3 does not "by definition" depend on the training data and thus on the sample size. However, the authors proposed a bound on this object that depends on the training size. This just doesn't make sense.

---

> > ### Author Response · Authors · 2023-11-23
> > **Response to Reviewer tc4X**
> >
> > Thanks for your comment! After decomposing and amplifying the gap in Definition.3, we can obtain it's upper bound in Eq. 4 and we indeed only focus on this upper bound in our work,  and carry out further derivation about introducing the information-theoretic quantity. Thus we can see that there exists an optimal hypothesis  $\hat{h}$ in terms of the true empirical risk in Eq. 4. Besides, it can be found that in Eq. 4, we study the participation gap with respect to $\hat{h}$ in fact.

---

### Official Review · Reviewer_kMKS · 2023-10-31

**Soundness:** 2 fair
**Presentation:** 2 fair
**Contribution:** 2 fair
**Rating:** 3
**Confidence:** 4

**Summary:**

This paper addresses the out-of-distribution generalization challenge in federated learning through a client-sampling approach inspired by information theory. Specifically, the authors propose to minimize the “self-information weighted empirical risk” function whose generalization bound leads to two client sampling strategies. These strategies aim to maximize the cross entropy between a participating client and a new client, assuming that novel clients very different data distributions. Empirical results suggest that the proposed sampling method works better than a range of baselines.

**Strengths:**

The paper presents a nice and clear definition of the new risk functions (Definitions 1 and 2).
The information-theoretic generalization bounds (Theorems 1 and 2) are presented well and explained with useful remarks.
In particular, the authors show that, from the client participation perspective, the bounds can be minimized by careful client sampling and setting the right the participation weights \alpha_i.

**Weaknesses:**

- There are a lot of unaccounted-for notations in this paper, beginning especially at Section 4. For example, the authors have not sufficiently explained what $F_i(.)$ and $w$ are in Assumption 3. Is $F_i(.)$ the usual empirical risk or information-theoretic version? Similarly, what is the local gradient $g_i^t$ with respect to?
- This notational ambiguity makes me unable to understand Algorithm 2 fully. At line 5, especially, what local loss function does client i try to minimize?
- In Section 4.2.2, again, I do not fully understand what the convex hull is with respect to.
- Empirically speaking, it is quite rare to update clients’ parameters through only one round of gradient descent as the authors propose in Algorithm 2, due to the cost of communication relative to local computation.

**Questions:**

- The authors aim to optimize the participation-dependent term in the generalization bound. However, the relationship between this problem and the formulations in (11) and (12) and insufficiently clear. For example, what do the authors mean by the claim that cosine similarity is “suitable for FL”? Similarly, I do not see any proof for the equivalence of this optimization problem and that in (12).

---

> ### Author Response · Authors · 2023-11-13
> **Response to Reviewer kMKS**
>
> Thanks very much for your detailed comments! The following are our responses to the comments accordingly:
>
> **Weaknesses**:
> * **Unaccounted-for notations in this paper.** Thank you for reminding us! In this section,  $F_i$ denotes the usual empirical risk and $g_i^t$ indicates the local gradient with respect to the usual empirical risk $F_i$. It is important to note that the self-information weighted expected risk and the corresponding information-theoretic generalization gap considered in this paper provide insights to guide the algorithm design for improving the out-of-distribution generalization performance of federated learning.  This is because the oracle probability $P(z)$ of outcomes is unknown in practice.
>
> * **Notational ambiguity in Alg.2.** Thank you once again for bringing this to our attention! The local loss function in Line 5 of Algorithm 2 refers to the standard loss function commonly used in FL, i.e., the usual mini-batch loss $\frac{1}{|B_i|}\sum_j \ell(z_j), z_j \in B_i, B_i \subset S_i$, where $B_i$ is the mini-batch of i-th local dataset $S_i$.
>
>
> * **About the convex hull.** The detailed procedure of the proposed convex hull-based client selection method is described in the supplementary material and we will discuss it in this response below.
> The formal work process of the proposed convex hull construction-based client selection  is outlined as follows. a) The server initiates the execution of the quickhull algorithm to construct the convex hull of the local gradients stored in the gradient table. b) Once  the vertices of the considered point set, which comprises
>  all the stored gradients are identified, the server proceeds to select the corresponding clients whose gradients are located on these discovered vertices. These selected clients are then required to participate in the subsequent round of FL.
>
> * **One round of gradient descent in local update.** The derived generalization bound is based on the uniform convergence-type generalization bound used in the classic study of federated generalization in (Hu et al., 2023). This theoretical technique focuses on the algorithm-independent generalization bound, which means that the derived generalization bound is not based on one specific algorithm, such as FedSGD or FedAvg. In other words, the information-theoretic generalization framework can be applied in one-round or multi-round local optimization settings since it is based on the algorithm-independent generalization bound.
>
> **Questions: The relationship between optimizing the participation dependent term and the formulations in (11) and (12):** The cross entropy term $\sum_{j}H(P_i,P_j), i\in\mathcal{I}_t, j \in \mathcal{I}_p \setminus \mathcal{I}_t $ plays a crucial role in the generalization bound presented in Theorem 2. It indicates that the distribution discrepancy between the selected data sources and the unselected data sources can significantly impact the out-of-distribution generalization performance of federated learning.  Furthermore, it is worth noting that in the field of federated learning, the distance between different local gradients $\Vert \nabla F_i-\nabla F_j\Vert^2, \forall i,j \in \mathcal{I}_p$ can be considered as an approximate metric of distribution discrepancy, as discussed in the study (Zou et al, 2023). This distance can be further expressed as $ \Vert\nabla F_i-\nabla F_j\Vert^2=\Vert \nabla F_i\Vert^2-2<\nabla F_i,\nabla F_j>+\Vert \nabla F_j\Vert^2$. In the context of federated learning, optimizing the cosine similarity $\frac{<\nabla F_i,\nabla F_j>}{\Vert \nabla F_i\Vert\Vert \nabla F_j\Vert}$ provides a more stable approach (Zeng et al, 2023).
>
>
>
> Xiaolin Hu, Shaojie Li, and Yong Liu. Generalization bounds for federated learning: Fast rates, unparticipating clients and unbounded losses. In International Conference on Learning Representations, 2023.
>
> Yinan Zou, Zixin Wang, Xu Chen, Haibo Zhou, and Yong Zhou. Knowledge-guided learning for transceiver design in over-the-air federated learning. IEEE Transactions on Wireless Communications, 22(1):270–285, 2023. doi: 10.1109/TWC.2022.3192550.12
>
> Dun Zeng, Xiangjing Hu, Shiyu Liu, Yue Yu, Qifan Wang, and Zenglin Xu. Stochastic clustered
> federated learning, 2023.

---

### Official Review · Reviewer_pRU9 · 2023-11-01

**Soundness:** 2 fair
**Presentation:** 2 fair
**Contribution:** 2 fair
**Rating:** 6
**Confidence:** 4

**Summary:**

This work studies generalization of Federated Learning in the non-IID setting. The main result is a generalization bound of a so-called 'self-information weighted expected risk', i.e., the expected risk weighted by the empirical entropy of data. Based on this result, the paper proposes an entropy-weighted aggregation method and client selection methods to improve FL training. On EMNIST-10,  Shakespeare, and CIFAR-10 datasets, the proposed methods are observed to improve Out-Of-Distribution generalization.

**Strengths:**

1. This paper defines a seemingly new objective called Information theoretic-generalization gap. This objective captures the uncertainty in data distributions and client dropout, which are absent in the classical notions of generalization error.

2. The theoretical framework appears to potentially have relevant, and positive impact to FL practice, as it leads to some new methods for gradient aggregation and client selection. It is good to see that the authors perform experiments on their algorithms and compare with baseline.

3. The paper is mostly clear and easy to read.

**Weaknesses:**

1. First of all, it is unclear what it the meaning of bounding the self-information weighted expected risk. In practice, we care about the accuracy over the generalization dataset, which is different from the information-weighted risk. Therefore, the fundamental question is: why bounding the information theoretic-generalization gap means good generalization accuracy? I didn't found in this paper an answer to this question or any solid explanations on this connection.

2. The upper bound in Theorem 1 lacks examples to explain each term in concrete cases. Given its current form, it is nearly impossible to compare with existing bounds and reasoning about tightness.

3. VC dimension is usually too large for modern neural networks to make it a useful complexity measure. Unfortunately, theorem 1 involves VC dimension.

4. The experiment section seems also quite limited in that only some simple and small-scale datasets are tested.

**Questions:**

1. What is the connection of information theoretic-generalization gap and generalization accuracy?

2. What are some examples to explain each term of the upper bound in Theorem 1, in concrete cases?

3. How to compare with existing bounds and reasoning about tightness on Theorem 1?

4. Speaking about OOD generalization, is there anywhere in the framework that captures the distribution shift? I thought something like mutual information should appear?

5. In Table 1, why is your methods' ID accuracy on CIFAR-10 worse than MaxSim, Power-of-Choice, and Full Sampling?

---

> ### Author Response · Authors · 2023-11-13
> **Response to Reviewer pRU9**
>
> We sincerely appreciate the supportive and constructive comments received! In the following sections, we will address the weaknesses and questions raised by the reviewer regarding our paper.
>
> **Weakness**:
>
> * **The meaning of bounding the self-information weighted expected risk.** The motivation behind defining the self-information weighted expected risk is described in Equation 1. To further support this concept, we will provide a concrete example. During the training stage, the target distributions are unknown. It is natural to assume that these unknown target distributions have greater uncertainty. For instance, we can consider a uniform distribution of discrete labels as an example. In such cases, it is reasonable to incorporate the self-information of outcomes to formulate the expected risk.  This is because the considered model $h\in\mathcal{H}$  may perform poorly on certain outcomes $z \in \mathcal{Z}$ with low probability in the training distributions, while these outcomes may have higher probabilities in the unknown target distributions. Here we provide a concrete example: Internet of Things (IoT) devices utilize spatially correlated data for federated learning, where the data collected by devices exhibits spatial heterogeneity. Devices participating in federated learning have limited data collected from certain regions, resulting in the possibility of the global model obtained through federated learning providing poorer services to devices that have collected a larger amount of data from that region.
> * **Concrete cases of each term in the upper bound in Theorem 1.** The first term $\sup_{h \in \mathcal{H}}L\Vert\hat{h}^*-h\Vert H(Z^{\mathcal{I}})$ in the upper bound in Theorem 1 reflects the discrepancy between the optimal global model $\hat{h}^*=\arg\inf_{h}\mathcal{L}_{\mathcal{I}_p}(h)$, considering self-information of outcomes, and other models $h$. Therefore, this first term essentially captures the impact of model complexity on the considered generalization gap.  It is also impacted by the joint entropy $H(Z^{\mathcal{I}})$ of all the data sources,  indicating that the complexity of hypothesis space $\mathcal{H}$ and data sources $Z^{\mathcal{I}}$ affect the generalization. The second term $3bH(Z^{\mathcal{I}}) -b \sum_i H(Z_i), i \in \mathcal{I}_p$ measures the difference between the joint entropy of all the data sources and the sum of information entropy of participating data sources. The insight behind this crucial term is that reducing the uncertainty of all the data sources, including unknown target data sources, and increasing the information of participating data sources will help decrease the considered generalization bound. The third term, which incorporates model complexity (VC dimension) and sample complexity, is commonly found in generalization theory (Hu et al., 2023).
>
>
> * **VC dimension used in the proposed generalization bound.** First of all, we follow the setting of the classic generalization study of federated learning (Hu et al., 2023), which considers unparticipating clients. Therefore, we directly employ the techniques they used to bound the generalization gap, namely the VC dimension. Besides, alternative metrics such as Rademacher complexity and covering number can also be used in the uniform convergence-type generalization bound discussed in this paper, refining the generalization bound further. However, this paper’s focus is on designing algorithms to improve the out-of-distribution generalization of federated learning, inspired by the proposed information-theoretic generalization framework. Enhancing traditional generalization techniques to refine the generalization bound is beyond the scope of this paper.
> * **Experimental results.**  Experiment setup of this paper follows previous works on federated generalization studies (Yuan et al, 2021; Hu et al, 2023). We believe the key points of this paper are fully considered by experiments. It will be appreciated if the reviewer provides direct requirements of the further experiment study. We are also willing to conduct further experiments if suggested.
>
>
>
> Xiaolin Hu, Shaojie Li, and Yong Liu. Generalization bounds for federated learning: Fast rates, unparticipating clients and unbounded losses. In International Conference on Learning Representations, 2023.
>
> Honglin Yuan, Warren Morningstar, Lin Ning, and Karan Singhal. What do we mean by generalization
> in federated learning? arXiv preprint arXiv:2110.14216, 2021.

---

> > ### Author Response · Authors · 2023-11-13
> > **Response to Reviewer pRU9**
> >
> > **Questions**:
> > * **Connection of information theoretic-generalization gap and generalization accuracy.**  As mentioned earlier, the proposed information-theoretic generalization gap is formulated by considering the self-information of outcomes. Therefore, it is well-suited for situations where the target distribution is unknown during the training stage. Consequently, if the information-theoretic generalization gap is decreased, the out-of-distribution generalization accuracy tends to decrease as well.
> > * **Examples to explain each term of the upper bound in Theorem 1, in concrete cases.**  We have provided detailed examples in the weaknesses section of this comment.
> > * **About how to compare with existing bounds and reasoning about tightness on Theorem 1.** The generalization bound proposed in this paper is derived based on the considered self-information weighted expected risk. The theoretical framework of this paper differs from other studies on federated generalization. Therefore, comparing the proposed generalization bounds with existing bounds is beyond the scope of this paper. Our focus is on developing algorithms inspired by the proposed information-theoretic generalization framework.
> > * **The framework captures the distribution shift.** In fact, both the difference between the joint distribution of the sum of information entropy of participating data sources $3bH(Z^{\mathcal{I}})-\sum_i H(Z_i), i \in \mathcal{I}_p$ in Theorem 1 and the cross entropy $ \sum_j H(P_i ,P_j), i\in \mathcal{I}_t, j \in \mathcal{I}_p \setminus \mathcal{I}_t$ in Theorem 2 capture the distribution shift. The mutual information $I( Z^{\mathcal{I}_t};Z^{\mathcal{I}_p \setminus \mathcal{I}_t})$ also affects the generalization bound. Please refer to the proof of Theorem 2 on page 19 of the supplementary material for more details.
> > * **The reason of proposed methods' ID accuracy on CIFAR-10 show worse.** The algorithm designed in this paper is based on the proposed information-theoretic generalization framework. The core idea is to consider the information entropy of individual data sources and the distribution discrepancy between different sources to achieve distribution diversification. However, baseline methods such as MaxSim, Power-of-Choice, and Full Sampling aggregate local models from data sources with lower information content and smaller distribution differences. Consequently, when these aggregated global models are tested on a target domain that shares the same distribution as the source domain, they tend to perform better, resulting in lower ID accuracy. However, it is important to note that the focus of this paper is to design methods for improving the out-of-distribution generalization performance of federated learning by considering the proposed information-theoretic generalization framework. Therefore, the focus is on the OOD accuracy rather than the ID accuracy.

---

> > > ### Comment · Reviewer_pRU9 · 2023-11-23
> > >
> > > Thank you for the response. I have no further questions and have raised my score.

---

> > > > ### Author Response · Authors · 2023-11-23
> > > > **Response to Reviewer pRU9**
> > > >
> > > > Thank you very much for your valuable comments once again! We sincerely appreciate that you have raised your score!

---

### Official Review · Reviewer_poBc · 2023-11-12

**Soundness:** 3 good
**Presentation:** 3 good
**Contribution:** 3 good
**Rating:** 5
**Confidence:** 4

**Summary:**

This paper considered the problem of generalization performance in federated learning (FL) with non-i.i.d. data and partial client participation. The authors proposed an information-theoretic framework for FL that quantifies the generalization error by evaluating the information entropy of local distributions and discerning discrepancies across these distributions. Based on their derived generalization error bounds, the authors proposed a weighted aggregation approach and two client selection strategies. The authors also conducted numerical experiments to verify their proposed methods.

**Strengths:**

1. The authors focused on the divergence between the training distributions of participating clients and the testing distributions of the non-participating clients, which is less studied in the literature.

2. This paper has good theoretical depth and provides interesting insights with information-theoretic generalization error bounds.

**Weaknesses:**

1. The tightness of the information-theoretic generalization error bound is unknown. Thus, the weighted aggregation and client selection strategies based on the generalization error bound are unclear. Also, several notions in the derived information-theoretic generalization error bound are unclear.

2. The numerical experimentations may be inadequate.

**Questions:**

1. It's unclear how tight the proposed joint self-information generalization error bounds in Theorems 1 and 2 are. Also, it is known in the literature that the VC-dimension-based generalization error bound, which is also used in Theorems 1 and 2, could be loose. Could the authors provide corresponding lower bounds for the proposed joint self-information-based generalization error bounds to show the tightness of the upper bounds? This paper could benefit tremendously from such insights and provide theoretical guarantees for the subsequent aggregation weighting and client selection strategies.

2. In Definition 3, the information theoretic-generalization gap is based on the $\hat{h}^*$, which is the optimal model parameter corresponding to the proposed self-information weighted expected risk. A similar notion of $\hat{h}_t^*$ is used for the selected participating clients in Theorem 2. However, in practice, such optimal model parameters are rarely found due to the non-convexity of the problem. Instead, the model parameters in use are highly dependent on the problem setting (e.g., non-convexity, smoothness, etc.) and the optimization methods (e.g., a large number of FedAvg-type variants) and the associated hyper-parameters (e.g., learning rates, batch sizes, etc.). Thus, the generalization error bounds derived in Theorems 1 and 2 based on the optimal parameters might not be very meaningful. Could the authors further characterize generalization error bounds for commonly used FL algorithms (e.g., FedAvg-type)?

3. The numerical experiments in this paper are largely based on CNN EMNIST-10 and CIFAR-10, which are considered relatively simple in the literature. Could the authors conduct more comprehensive experiments and evaluate their proposed weighted aggregation and client selection strategies with more challenging models (e.g., larger ResNet-type models) and datasets?

---

> ### Author Response · Authors · 2023-11-13
> **Response to Reviewer poBc**
>
> Thanks for the detailed comments and reminders presented by the reviewer. We would like to respond to the weaknesses and questions to support our paper.
>
> * **About the tightness of the information-theoretic generalization error bound.** Our paper follows the same setting as the classic study of federated generalization by Hu et al. (2023). In this setting, there are unparticipating clients in FL, while the global model trained by participating clients still needs to provide service for these unparticipating clients. Therefore, we directly utilize the theoretical techniques used in their study, specifically the uniform convergence-type generalization bounds.  In these generalization bounds, various metrics are used to measure the complexity of the model. While VC dimension is one commonly used metric, it is important to note that alternative metrics such as Rademacher complexity or covering number can also be employed to refine the proposed generalization bound. However, exploring these alternative metrics falls beyond the scope of our paper. Our primary focus is on designing algorithms to improve the out-of-distribution generalization performance of FL, drawing insights from the proposed information-theoretic generalization framework. We firmly believe that the information entropy term in Theorem 1 and the cross-entropy term in Theorem 2 can provide valuable insights towards achieving this goal.
>
> * **About algorithm-dependent generalization bound.** Thank you very much for your valuable comment. Indeed, the factors mentioned in your comment, such as the optimization method, the problem setting (e.g., Lipschitz constant of the loss function), and related hyperparameters, can be studied using important techniques in generalization analysis known as “algorithm-dependent generalization bounds”. These techniques include algorithmic stability-based generalization bounds, PAC-Bayes-based generalization bounds, and others. However, the fundamental framework of generalization analysis employed in this paper is the uniform convergence-based generalization bound, which is algorithm-independent. In this framework, we focus on incorporating the self-information weighted expected risk and leveraging it to derive the corresponding generalization bound. We then design methods for improving the out-of-distribution generalization performance of FL motivated by our theoretical findings, which is the primary scope of our paper. In other words, deriving the algorithm-dependent generalization bound is beyond the scope of this paper. Furthermore, since the proposed generalization bound is algorithm-independent, it is well-suited for various federated optimization algorithms, such as FedAvg.
>
> * **About the numerical experimentations.** The experimental setting of this paper aligns with the existing theoretical works on federated generalization (Yuan et al., 2021; Hu et al., 2023). In their studies, Yuan et al. utilize the EMNIST and CIFAR datasets to verify their proposed two-level generalization framework, while Hu et al. employ the EMNIST dataset to validate their proposed uniform convergence-type generalization bound. In addition to these datasets, we also consider the Shakespeare dataset, which is commonly used in NLP, to evaluate the effectiveness of our proposed algorithms and to provide empirical support for our theoretical framework. By including this dataset in our experiments, we ensure that the core aspects of our paper are thoroughly examined, and the experimental results serve as strong evidence to support our proposed theoretical framework.
>
> Honglin Yuan, Warren Morningstar, Lin Ning, and Karan Singhal. What do we mean by generalization in federated learning? arXiv preprint arXiv:2110.14216, 2021.
>
> Xiaolin Hu, Shaojie Li, and Yong Liu. Generalization bounds for federated learning: Fast rates, unparticipating clients and unbounded losses. In International Conference on Learning Representations, 2023.

---

### Author Response · Authors · 2023-11-17
**Summary of the revision.**

We have diligently refined and revised our paper based on the valuable comments provided by the reviewers. The key revisions can be summarized as follows:



**Motivation of the proposed self-information weighted expected risk:** We have further added more motivating practical examples to strengthen the rationale behind the definition of the self-information weighted expected risk proposed in our paper.

**Further clarification of the remark:** We have improved Remark 1 by providing detailed explanations for each term in Theorem 1, enhancing the clarity and comprehensibility of the remark.

**Further explanation about the proposed algorithms:** We have included an explanation of why the cosine similarity can be employed as a proxy to enhance the generalization performance of FL.


**Definitions:** We have refined the introduction of the proposed “semi-empirical self-information weighted expected risk” and elucidated the connection between this definition and a similar definition proposed in (Hu et al., 2023).

**Notations:** We have meticulously reviewed the notation and addressed minor issues throughout the paper, with particular attention to the methods section.

---

### Meta-Review · Area_Chair_LUsN · 2023-12-05

**Metareview:**

This paper attempts to use information theory to formulate generalization errors of federated learning when the nature of clients is heterogeneous. Specifically, the paper defines risks to be aggregated by weighting based on the information theory and proposes upper bounds for generalization errors and methods using these risks. The problem with this paper is that it is limited to the analysis of the defined risks and does not justify the risks themselves or their advantages. Another negative factor is that there is little technical novelty in the analysis and the contribution of the theory is limited. It is also noted that the mathematical description is loose, and further improvement is desirable.

**Justification For Why Not Higher Score:**

The weaknesses of this paper are clearly identified and there is consensus among the reviewers that there is room for improvement.

**Justification For Why Not Lower Score:**

N/A

---

### Decision · Program_Chairs · 2024-01-16

Reject